# Evolutionary genomics of epidemic visceral leishmaniasis in the Indian subcontinent

Hideo Imamura[1†], Tim Downing[2,3†‡], Frederik Van den Broeck[1†], Mandy J Sanders[2], Suman Rijal[4], Shyam Sundar[5], An Mannaert[1], Manu Vanaerschot[1§], Maya Berg[1], Géraldine De Muylder[1], Franck Dumetz[1], Bart Cuypers[1], Ilse Maes[1], Malgorzata Domagalska[1], Saskia Decuypere[1,6], Keshav Rai[4¶], Surendra Uranw[4], Narayan Raj Bhattarai[4], Basudha Khanal[4], Vijay Kumar Prajapati[5**], Smriti Sharma[5], Olivia Stark[7], Gabriele Schönian[7], Harry P De Koning[8], Luca Settimo[8,9], Benoit Vanhollebeke[10], Syamal Roy[11], Bart Ostyn[12], Marleen Boelaert[12], Louis Maes[13], Matthew Berriman[2], Jean-Claude Dujardin[1,13*], James A Cotton[2*]

[1]Department of Biomedical Sciences, Institute of Tropical Medicine, Antwerp, Belgium; [2]Wellcome Trust Sanger Institute, Hinxton, United Kingdom; [3]School of Maths, Applied Maths and Statistics, National University of Ireland Galway, Galway, Ireland; [4]BP Koirala Institute of Health Sciences, Dharan, Nepal; [5]Department of Medicine, Institute of Medical Sciences, Banaras Hindu University, Varanasi, India; [6]Telethon Kids Institute, University of Western Australia, Perth, Australia; [7]Institut für Mikrobiologie und Hygiene, Charité Universitätsmedizin Berlin, Berlin, Germany; [8]Institute of Infection, Immunity and Inflammation, College of Medical, Veterinary and Life Sciences, University of Glasgow, Glasgow, United Kingdom; [9]Department of Chemistry and Chemical Biology, Northeastern University, Boston, United States; [10]Laboratory of Molecular Parasitology, Université Libre de Bruxelles, Gosselies, Belgium; [11]Department of Infectious Diseases and Immunology, Council of Scientific and Industrial Research, Indian Institute of Chemical Biology, Kolkata, India; [12]Department of Public Health, Institute of Tropical Medicine, Antwerp, Belgium; [13]Department of Biomedical Sciences, Faculty of Pharmaceutical, Biomedical and Veterinary Sciences, University of Antwerp, Antwerp, Belgium

*For correspondence: JCDujardin@itg.be (JCD); james. cotton@sanger.ac.uk (JAC)

[†]These authors contributed equally to this work

Present address: [‡]School of Biotechnology, Dublin City University, Dublin, Ireland; [§]Department of Microbiology and Immunology, Columbia University College of Physicians and Surgeons, New York, United States; [¶]West Bengal State University, Kolkata, India; [**]Department of Biochemistry, Central University of Rajasthan, Ajmer, India

Competing interests: The authors declare that no competing interests exist.

**Abstract** *Leishmania donovani* causes visceral leishmaniasis (VL), the second most deadly vector-borne parasitic disease. A recent epidemic in the Indian subcontinent (ISC) caused up to 80% of global VL and over 30,000 deaths per year. Resistance against antimonial drugs has probably been a contributing factor in the persistence of this epidemic. Here we use whole genome sequences from 204 clinical isolates to track the evolution and epidemiology of *L. donovani* from the ISC. We identify independent radiations that have emerged since a bottleneck coincident with 1960s DDT spraying campaigns. A genetically distinct population frequently resistant to antimonials has a two base-pair insertion in the aquaglyceroporin gene LdAQP1 that prevents the transport of trivalent antimonials. We find evidence of genetic exchange between ISC populations, and show that the mutation in LdAQP1 has spread by recombination. Our results reveal the complexity of *L. donovani* evolution in the ISC in response to drug treatment.

**eLife digest** The parasite *Leishmania donovani* causes a disease called visceral leishmaniasis that affects many of the world's poorest people. Around half a million new cases develop every year, but health authorities lack safe and effective drugs to treat them. Up to 80% of these cases occur in the Indian subcontinent, where devastating epidemics have occurred in the last decades.

One reason these epidemics continue to occur is that the parasites develop genetic mutations allowing them to adapt to and resist the drugs used to kill them. As there are few existing drugs that can kill *L. donovani*, it is crucial to understand how drug resistance emerges and spreads among parasite populations.

Imamura, Downing, Van den Broeck et al. have now investigated the history of visceral leishmaniasis epidemics by characterising the complete genetic sequence – or genome – of 204 *L. donovani* parasite samples. This revealed that the majority of parasites in the Indian subcontinent first appeared in the nineteenth century, matching the first historical records of visceral leishmaniasis epidemics.

The genomes show that most of the parasites are genetically similar and can be clustered into several closely related groups. These groups first appeared in the 1960s following the end of a regional campaign to eradicate malaria. The most common parasite group is particularly resistant to drugs called antimonials, which were the main treatment for leishmaniasis until recently. These parasites have a small genetic change that scrambles most of a protein known to be involved in the uptake of antimonials.

Parasites may also be able to develop resistance to drugs through additional mechanisms that allow them to produce many copies of the same gene. These mechanisms could allow the parasites to rapidly adapt to new drugs or changes in the populations it infects. The work of Imamura et al. looks only at parasites isolated from patients then grown in the laboratory, so further research is now needed to explore how variable the *Leishmania* genome is in both of the parasite's hosts: humans and sandflies.

Imamura et al.'s study reveals how *L. donovani* has spread throughout the Indian subcontinent in fine detail. The genome data can be used to create simple molecular tools that could form an "early warning system" to track the success of disease control programs and to determine how well the current drugs are working.

## Introduction

Parasites of the *Leishmania donovani* species complex cause visceral leishmaniasis (VL), the most severe presentation of leishmaniasis that is usually fatal if untreated. There are probably between 200,000 and 300,000 VL cases annually (*Alvar et al., 2012*), leading to as many as 50,000 deaths per year (*den Boer et al., 2011*; *Lozano et al., 2012*). VL is widespread in both the New and Old Worlds (*Pigott et al., 2014*), but as much as 80% of the global VL burden occurs in the Indian sub-continent (*Alvar et al., 2012*). Recent intensified control efforts have led to a notable decline in cases (*Chowdhury et al., 2014*) but the problem is not yet eliminated. VL is a key neglected tropical disease, affecting the poorest regions of the world and the poorest communities within these regions (*Boelaert et al., 2009*). VL was first reported in the Indian sub-continent (ISC) in the 1820s, but initially confused with malaria until the discovery of *L. donovani* in 1903 (*Gibson, 1983*). Although VL was nearly eliminated from the ISC in the 1960s (*Thakur, 2007*) by antimalarial spraying campaigns with DDT, it re-emerged in 1977 and has caused several subsequent major epidemics (*Dye and Wolpert, 1988*). Widespread chemotherapy for VL in the region has been ongoing since the 1820s, initially with quinine and other drugs, followed by extensive use of the trivalent antimonial $Sb^{III}$ (1915) and compounds of the less toxic pentavalent $Sb^{V}$ (1922) such as sodium stibogluconate (SSG), and since 2005 with miltefosine (MIL) that is freely supplied through a government-subsidized control program. The parasite developed resistance to both $Sb^{III}$ and $Sb^{V}$, and after ten years of clinical use there has been a notable decline in MIL efficacy (*Rijal et al., 2013*; *2007*; *Sundar et al., 2012*).

*Leishmania* parasites can re-shape their genome rapidly in vitro in response to stress (*Leprohon et al., 2009*), suggesting structural variation is an important feature by which they can

rapidly adapt to changing environmental conditions and drug pressure. However, there is little data on the diversity of clinical *Leishmania* populations or how they evolve during treatment. While an extensive literature has made use of molecular methods to study the population genetics of *Leishmania* (e.g. *Alam et al., 2009*; *Lukes et al., 2007*; *Mauricio et al., 2006*; *Schonian et al., 2008*), existing genetic markers have relatively poor resolution, and in particular *L. donovani* within the ISC show very little genetic differentiation based on these approaches (*Alam et al., 2009*; *Downing et al., 2012*). Whole-genome sequence data has the potential to show significant population structure within the ISC, and also allows us to identify changes in genome structure.

Here we report the genome sequences of 204 *L. donovani* isolates (*Figure 1*, *Supplementary file 1*), obtained from VL patients between 2002 and 2011 from regions in Nepal (N=98), India (N=98) and Bangladesh (N=8) that represent the epicentre of the on-going VL epidemic in the ISC (*Figure 1a*).

## Results

Calling variants against a reference genome assembly for a Nepalese *L. donovani* strain (BPK282/0cl4; *Figure 2—figure supplement 1*), we identify three divergent genetic lineages circulating in this region (*Figure 2b*): a core group of 191 closely related parasites found in the highly endemic

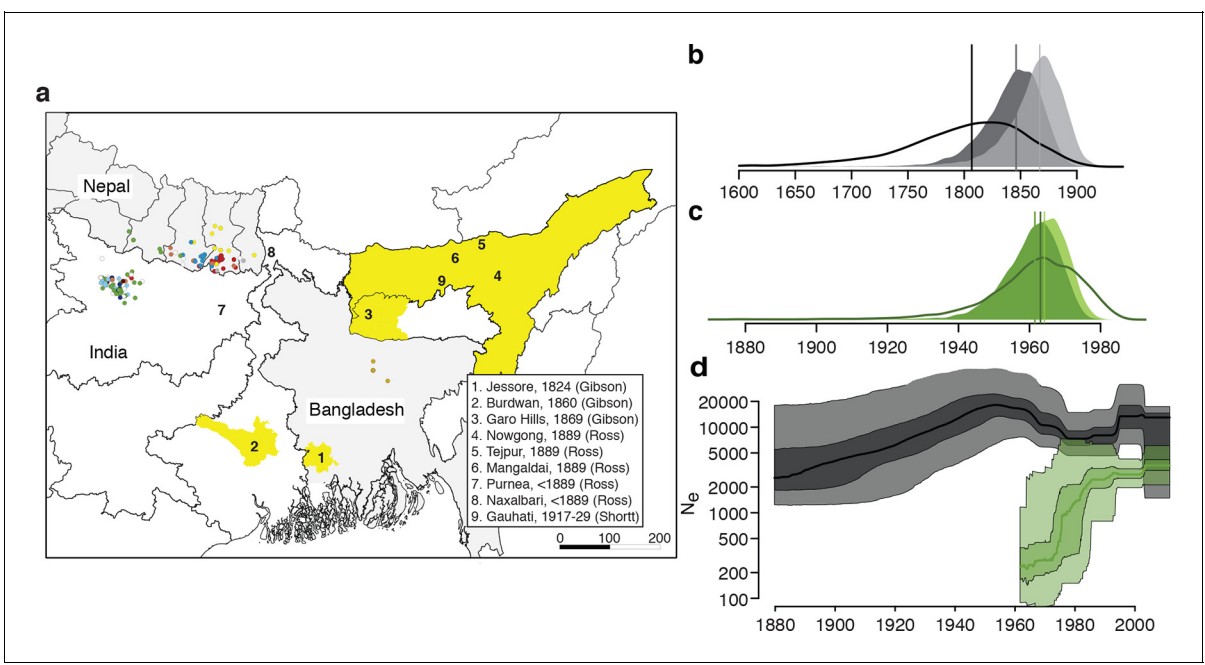

**Figure 1.** History and geography of Indian subcontinent *L. donovani*. (a) Location of the patients from which the 204 *L. donovani* genomes were isolated, and of historical Kala-Azar outbreaks. Genetic groups of the parasite isolates are indicated by the colour of the dots representing them, matching those in *Figure 2a,c*. Sampling dates and locations are summarised in *Figure 1—figure supplement 1*, and detailed information about each strain including GPS coordinates are given in the source data file. Citations are to historical primary literature reviewed and cited in (*Gibson, 1983*). Posterior probability distributions of estimated ages for the oldest split in (b) the main population in Bihar and Nepal and (c) the ISC5 group associated with Sb resistance. Dark shading shows estimates under a strict molecular clock, light shading from relaxed molecular clock and lines show relaxed clock results with Bangladeshi and putative hybrid isolates included. (d) Estimated effective population size through time for ISC5 population (green) and the rest of the parasite population (black/grey). Lines show median of posterior distributions, dark and light shading cover 50% and 95% of the posterior density respectively. Dates for all splits on this phylogeny and other results of phylogeographic analysis are shown in *Figure 1—figure supplement 2*.

The following figure supplements are available for figure 1:

**Figure supplement 1.** Sampling of genetic groups.

**Figure supplement 2.** Full results of discrete-space, constant population size molecular clock Bayesian phylogeography analysis of core population.

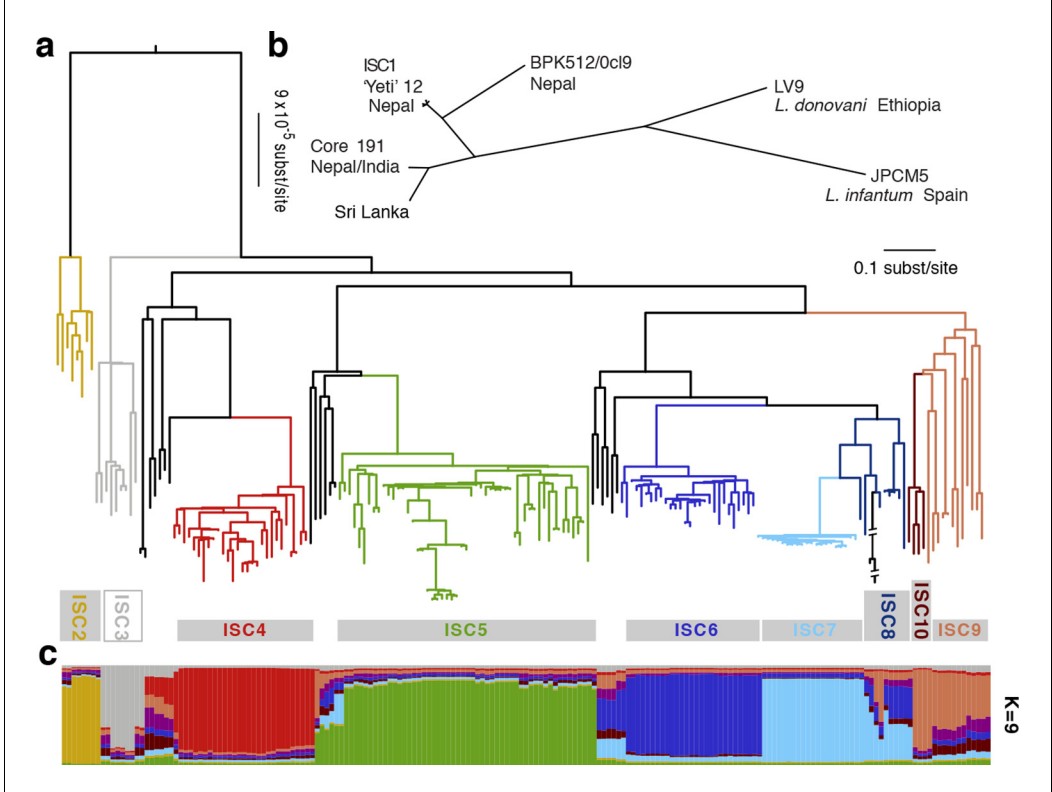

**Figure 2.** Genealogical history of *L. donovani* from the ISC. (**a**) Maximum-likelihood tree based on SNPs called for 191 strains (see *Figure 2—figure supplement 1*) from the core population in the Indian subcontinent. Samples are coloured by population assignment, with putative hybrid strains not clustered in the main groups in black. Further analysis confirms the hybrid ancestry of some of these isolates (*Figure 2—figure supplement 2*). (**b**) Unrooted phylogenetic network of the *L. donovani* complex based on split decomposition of maximum-likelihood distances between isolates described here, reference genome isolates and two published Sri Lankan isolates (*Zhang et al., 2014*). (**c**) Model-based clustering of 191 isolates from the core population reveals six discrete monophyletic groups, and some groups and other samples of less certain ancestry. Coloured bars show the fraction of ancestry per strain assigned to a given cluster, with colours assigned to the population most closely related to each cluster. More detailed population clustering analysis shows largely congruent results (*Figure 2—figure supplements 3* and *4*).

The following figure supplements are available for figure 2:

**Figure supplement 1.** Flowchart of SNP detection using COCALL.

**Figure supplement 2.** Haplotype networks for core population isolates.

**Figure supplement 3.** Haplotype similarity for core population isolates.

**Figure supplement 4.** Mosaic ancestry patterns in eight putative hybrid *L. donovani* isolates.

lowlands of all three countries, a small population of 12 Nepalese isolates found most frequently in the highlands (ISC1) and a single divergent Nepalese isolate (BPK512/0cl9) (*Downing et al., 2011*). These two main groups show fixed differences at 45,743 sites (*Supplementary file 2, table a*), and two previously sequenced Sri Lankan *L. donovani* isolates (*Zhang et al., 2014*) were more closely related to the core population (21,546 fixed differences) than to ISC1 (45,743 fixed differences). Parasites within each group show little SNP variation with only 5,628 variable sites in ISC1 and just 2,418 sites varying within the core population (*Supplementary file 2, table b*) and correspondingly few SNPs in protein-coding regions (*Supplementary file 2, table c*). Core population isolates differ at an average of 88.3 nucleotide sites with an average nucleotide diversity of 9.7 per Mb (*Supplementary file 2, table d*).

While a panel of microsatellite markers shows no variation between isolates from the core population (*Downing et al., 2012*), we reveal significant spatial and temporal genetic structure within this group despite this extremely low level of overall diversity (*Figure 2a,c*; *Supplementary file 2, table e*). Phylogenetic and clustering methods identify six congruent monophyletic groups (ISC2-7). Three other groups (ISC8-10) and 21 ungrouped isolates had more complex and less certain evolutionary histories (*Figure 2a,c*; *Supplementary file 2, table f*). Most of the ISC groups are present throughout our sampling window (2002–2011), and many are present in both India and Nepal (*Figure 1—figure supplement 1*). There are some exceptions: ISC7 represents a recent radiation (first observed in 2006) with almost no diversity (20 unique SNPs; *Supplementary file 2, table g*; π=1.8 per Mb) and is restricted to India, while ISC6 is an older and more diverse group restricted to Nepal (π=12.2 per Mb). We observe subsequent evolution within some groups: ISC5 is distinguished from other groups by just 32 SNP sites (*Supplementary file 2, table h*), but contains a subgroup with multiple novel SNPs and lower somy (*Supplementary file 2, table i*).

Bayesian phylogenetic models in an explicit temporal and spatial framework revealed that the core population diverged in the mid 19[th] century (*Figure 1b*), matching the dates of the earliest reports of large-scale VL outbreaks in the ISC (*Gibson, 1983*) and thus suggesting that modern lowland parasites descend from these early epidemics. Within the core population, the Indo-Nepalese population itself appeared around 1900 (*Figure 1—figure supplement 2*), almost certainly in India (0.89 posterior probability), matching the dates of the first reported outbreaks in Bihar (*Gibson, 1983*), more precisely in Purnea (*Figure 2d*). Most subsequent diversification is more recent, with many groups (ISC2 & ISC4-6) radiating from the 1960s (*Figure 1b*), coinciding with the end of the DDT spraying campaign. The estimated rate of migration from India to Nepal in the Core 191 group was significantly greater than that from Nepal to India, suggesting that India acts as a source population seeding the Nepalese epidemic (*Figure 1—figure supplement 2*).

A lack of linkage disequilibrium decay between SNP pairs with genomic distance in the core population ($r^2$~0.33 at 5–1,400 kb) reflects a lack of detectable recombination within the six main genetic groups (ISC2-7) across the entire genome (*Supplementary file 2, table j*). While the low number of SNPs varying within the core population limits our power to detect recombination, we find compelling evidence of hybridisation among eight of the samples not assigned to any of the ISC groups (*Figure 2—figure supplements 2–4*). The identity of these isolates as hybrids and our assignment of other isolates to groups is supported by allele-frequency based methods (f-statistics), which should be robust to gene flow between groups (*Supplementary file 2, tables e,f*) and population structure analysis based on haplotype sharing (*Supplementary files 2, tables k–n*). The four-allele test also confirms that recombination is largely restricted to these hybrids (*Supplementary file 2, table o*). These isolates appear to result from multiple independent recent hybridizations between distinct ancestors of either ISC5 and ISC6, ISC5 and ISC7, or ISC6 and ISC7 (*Figure 2—figure supplement 2*).

We detect extensive variation in the structure of these *L. donovani* genomes. Local copy-number variants (CNVs) cover ~11% of the genome. These include sporadic gene duplication, dynamic tandem gene array sizes (*Figure 3—figure supplement 1*) and long sub-telomeric amplifications/deletions, the latter generally spanning whole transcription units. While structural variation in *Leishmania* is often considered a transient adaptation, particularly to culture conditions in vitro, we find striking conservation of many CNVs across all core population groups here. Two multigenic intra-chromosomal duplicated regions (MAPK1 and H-locus; *Downing et al., 2011*) are present in variable numbers in all core population isolates but are absent in ISC1 (*Figure 3b,c*; *Figure 3—figure supplement 2*). Conserved heterozygous SNPs in both of these structural variants confirm that these regions have duplicated once and been maintained throughout the evolution of this population. All known genes on these duplicated regions are associated with virulence (MAPK1, ASS, sAcP; *Fernandes et al., 2013*; *Lakhal-Naouar et al., 2012*; *Wiese, 1998*) or drug resistance (*Brotherton et al., 2013*), indicating that extensive structural variation allows these parasites to alter local copy number in response to changing environments: both aneuploidy and CNV regulate gene expression (*Leprohon et al., 2009*). Most isolates are aneuploid (*Figure 3—figure supplement 3*), even excluding the generally tetrasomic chromosome 31, and almost all chromosomes show some variation in somy (*Figure 3a*). Aneuploidy ($r^2$=0.15, p=2.7x10$^{-118}$), CNVs ($r^2$=0.26, p=7.5x10$^{-218}$) and indels ($r^2$=0.30, p=2.1x10$^{-254}$) are significantly correlated with SNP variation in the core isolates, suggesting that these variants have appeared gradually during the evolution of the population in the

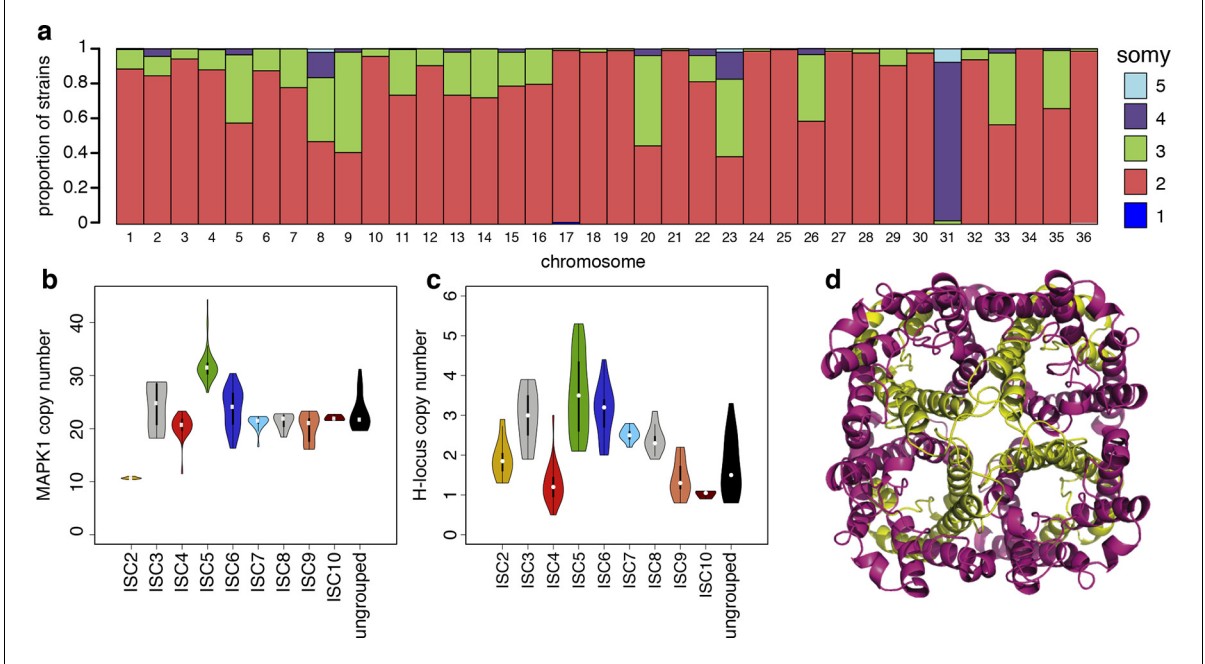

**Figure 3.** Structural variations in ISC *L. donovani*. (a) Stacked barplots per chromosome showing the proportion of ISC strains that are monosomic, disomic, trisomic, tretrasomic or pentasomic for the respective chromosome. A full breakdown of somy per strain is presented in *Figure 3—figure supplement 3*, and a complete catalogue of other structural variants in *Figure 3—figure supplement 1*. Violin plots showing the copy number of MAPK1 (b) and H-locus (c) per ISC group, except for ISC1 where these amplicons were absent. These amplicons are intra-chromosomal (*Figure 3—figure supplement 2*). (d) Tetrameric protein model of the transport protein aquaglyceroporin-1. The C-terminus part that is affected by the 2-nucleotide frameshift found in all ISC5 isolates is shown in magenta. Image was created using PyMOL version 1.50.04 (Schrödinger).

The following figure supplements are available for figure 3:

**Figure supplement 1.** Copy number variants in all 206 genomes.

**Figure supplement 2.** Copy number variation by intrachromosomal tandem duplication or extrachromosomal linear amplification in clinical isolate.

**Figure supplement 3.** Chromosome number variation in *L. donovani* in the ISC.

field. Most strikingly, we find two cases of recent epidemic expansions associated with major changes in aneuploidy and heterozygosity (*Figure 4*). Variation in somy can thus lead to changes in heterozygosity, which could allow selection to eradicate recessive deleterious mutations in the absence of recombination (*Roze and Michod, 2010*).

We find no statistically significant association between any individual SNP or structural genetic variant and in vitro SSG resistance, or SSG or MIL treatment outcomes (*Supplementary file 2, tables p–r*), but the distribution of antimony susceptibility was uneven across different ISC populations (*Supplementary file 1*). 9 of 11 ISC5 samples tested were highly $Sb^V$-resistant and two out of four ISC5-ISC6/7 hybrids tested have intermediate levels of resistance. One variant – a two-bp insertion introducing a frameshift and premature stop codon in the aquaglyceroporin-1 gene (LdBPK_310030, AQP1) – is homozygous in all 52 ISC5 isolates (*Table 1*), and heterozygous in six hybrids between ISC5 and either ISC6 or ISC7. ISC5 isolates also share other genomic features – such as higher copy number of both the H-locus and MAPK1 amplicons (*Figure 3b,c*). The H-locus includes MRPA, a gene involved in the efflux of $Sb^{III}$ and associated with drug resistance (*Leprohon et al., 2009*). Other lines of evidence strongly link AQP1 with antimony resistance. While recent antimonial drugs such as SSG are compounds of pentavalent antimony ($Sb^V$), $Sb^V$ is thought to act mostly as a prodrug, being reduced to $Sb^{III}$ in both the macrophage phagolysosome (*Frézard et al., 2001*) and in the parasite itself (*Denton et al., 2004*; *Decuypere et al., 2012*). AQP1 is known to assist with $Sb^{III}$ uptake, both genetic and transcriptional changes at this locus have been associated with Sb

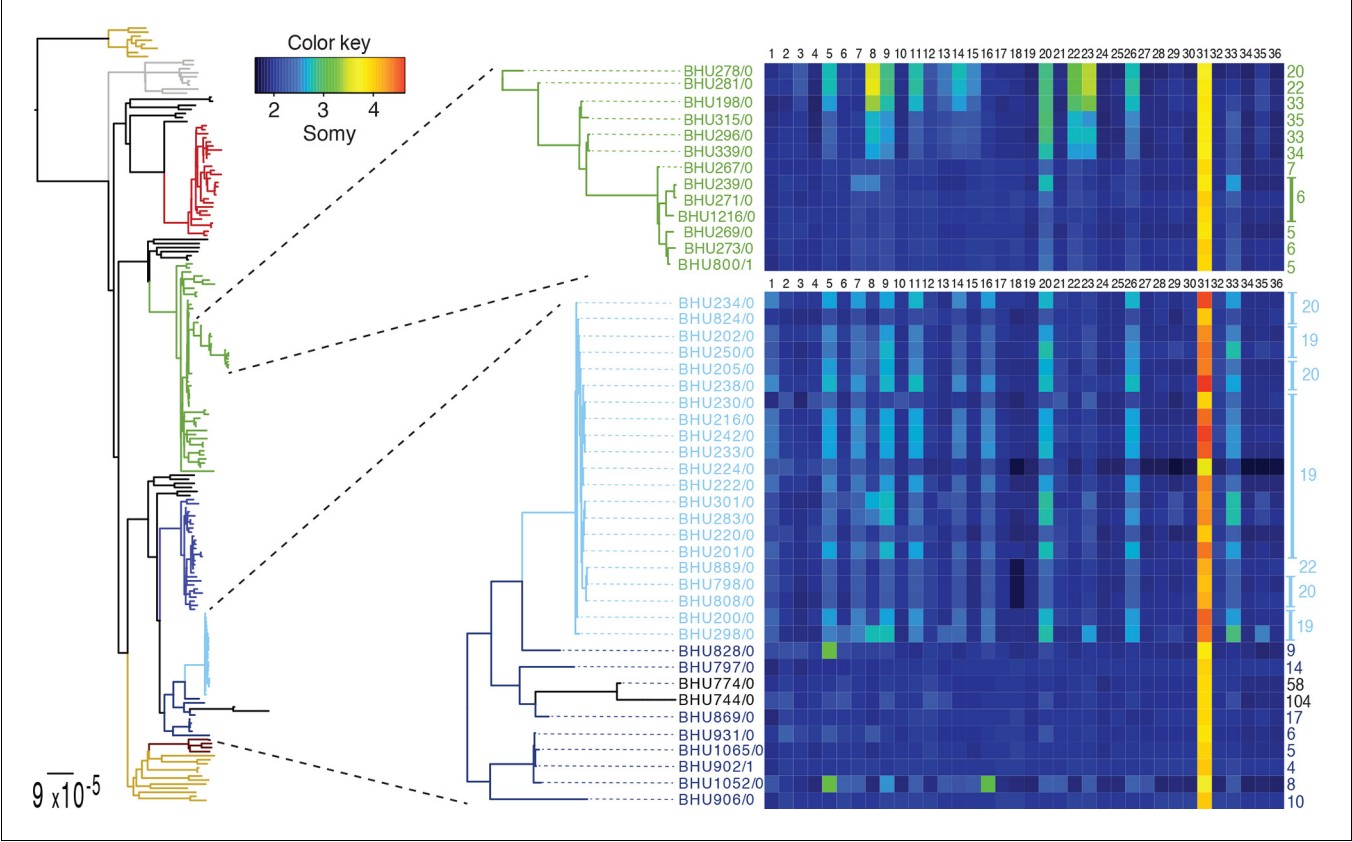

**Figure 4.** SNP heterozygosity and somy variation in two subclades. Two subclades show an expansion of polysomic strains from disomic ancestors (below) and an expansion of disomic strains from polysomic ancestors (above). Somy variation per chromosome (1–36; above heatmap) and the total number of heterozygote SNPs (right to heatmap) are shown for each individual strain.

resistance (*Gourbal et al., 2004*; *Monte-Neto et al., 2015*; *Mukherjee et al., 2013*; *Uzcategui et al., 2008*) and a homologous transporter is associated with drug resistance in trypanosomes (*Baker et al., 2012*). Recently, an AQP1 knockout line of *Leishmania major* was shown to be resistant to Sb[III] due to reduced uptake (*Plourde et al., 2015*). The truncated frameshift protein found in ISC5 is predicted to be incapable of forming a functional trans-membrane channel (*Figure 3d*). We find three other independent frameshifts in AQP1 gene in other antimony resistant isolates, including one in BPK181/12 (ISC6), an isolate taken from a patient following failure of ten months of antimony treatment that was absent in the pre-treatment isolate from the same patient (BPK181/ 0cl11, *Table 1*).

We propose that the AQP1 truncation is associated with antimonial resistance in the ancestor of ISC5, and has been transmitted to a group of hybrid parasites. The ISC5 lineage emerged following the end of the DDT campaigns but then proliferated quickly in the 1970s (*Figure 1c,d*), when antimonial dosage had to be doubled because of its declining efficacy. The persistence of this lineage beyond the era of Sb treatment perhaps reflects the increased fitness (*Vanaerschot et al., 2013*) of Sb-resistant parasites. These observations tally with a stronger signature of purifying selection on the ISC5 lineage, measured as a lower rate of derived allele accumulation compared to other ISC populations, which may be a consequence of higher historical exposure to drug stress (*Supplementary file 2, table s*). Sb resistance is also present in other genetic groups, with 4 out of 15 ISC4 lines tested in vitro being Sb[V]-R, indicating resistance has emerged independently and recently multiple times in ISC *L. donovani*, and that other genetic variants responsible for Sb[V] resistance must be present in this population. Indeed, other Sb[V] resistance mechanisms are known in this population: previous work has shown that two resistant strains from ISC4 (BPK087 and BPK190) show significantly decreased transcription of an AQP1 locus encoding a wildtype protein sequence

**Table 1.** Small indels. The first half of the table summarises the numbers and types of indels detected in each group. The second half of the table shows the proportion of samples within a cluster that share each group-specific coding-region indel.

| 1. Number of indels | ISC002 | ISC003 | ISC004 | ISC005 | ISC006 | ISC007 | ISC008 | ISC009 | ISC010 |
|---|---|---|---|---|---|---|---|---|---|
| Total number ofindels found within each group | 58 | 60 | 73 | 79 | 65 | 55 | 55 | 84 | 60 |
| Number of group-specific Indels shared by part of the strains of that group | 9 | 5 | 11 | 12 | 7 | 0 | 8 | 22 | 7 |
| Number of group-specific Indels shared by all the strains of that group | 6 | 1 | 3 | 3 | 0 | 0 | 0 | 0 | 2 |
| Number of group-specific Indels within coding regions | 0 | 1 | 2 | 3 | 4 | 0 | 1 | 3 | 1 |

2. Indels within coding region

| Gene ID | Gene product | Position | Type | ISC002 | ISC003 | ISC004 | ISC005 | ISC006 | ISC007 | ISC008 | ISC009 | ISC010 |
|---|---|---|---|---|---|---|---|---|---|---|---|---|
| LdBPK_310030 | Aquaglyceroporin | Ld31_0007774 | 2 | | | 0.04 | | | | | | |
| LdBPK_310030 | Aquaglyceroporin | Ld31_0007735 | 2 | | | | 1 | | | | | |
| LdBPK_310030 | Aquaglyceroporin | Ld31_0007662 | 1 | | | | | 0.04 | | | | |
| LdBPK_310030 | Aquaglyceroporin | Ld31_0008099 | 2 | | | | | 0.11 | | | | |
| LdBPK_291860 | Putative historie H2A | Ld29_0816454 | -2 | | | | | | | | 0.25 | |
| LdBPK_040410 | Conserved hypothetical protein | Ld04_0155491 | 3 | | | | | | | | 0.08 | |
| LdBPK_070540 | Conserved hypothetical protein | Ld07_0230487 | -3 | | | | | | | 0.12 | | |
| LdBPK_190080 | Conserved hypothetical protein | Ldl9_0015151 | 1 | | | | 0.04 | | | | | |
| LdBPK_261790 | Conserved hypothetical protein | Ld26_0651748 | 4 | | | | | 0.11 | | | | |
| LdBPK_301000 | Conserved hypothetical protein | Ld30_0311376 | -1 | | | | 0.02 | | | | | |
| LdBPK_310690 | Conserved hypothetical protein | Ld31_0241951 | -3 | | | | | 0.11 | | | | |
| LdBPK_332580 | Conserved hypothetical protein | Ld33_0995960 | 1 | | 0.04 | | | | | | | |
| LdBPK_366590 | Conserved hypothetical protein | Ld36_2473775 | -3 | | | | | | | | | 0.25 |
| LdBPK_110650 | hypothetical, unknown function | Ldll_0245832 | -3 | | 0.56 | | | | | | | |
| LdBPK_292330 | hypothetical, unknown function | Ld29_1008496 | -3 | | | | | | | | 0.08 | |

(*Decuypere et al., 2005*), and BHU764 combines a different indel mutation in AQP1 and reduced expression of MRPA, an efflux transporter of $Sb^{III}$ (*Mukhopadhyay et al., 2011*). The failure of any single resistance locus to sweep through this population may reflect the low level of gene flow and the presence of a large reservoir of untreated asymptomatic cases (*Ostyn et al., 2011*).

## Discussion

We have shown that genomic data can retrospectively unravel the evolution and epidemiology of this parasite population, and gain new insight into possible mechanisms of drug resistance against a background of extensive variation in genome structure. We report the first analysis of the structure and history of a *Leishmania* population, aligned with clinical and epidemiological records, enabled by the higher resolution of genome sequence data than other genotyping approaches. These data have allowed us to describe a mechanism of resistance to one of the most ancient drugs used in the

human pharmacopeia, antimonials, not only identifying a key locus, but also showing the epidemiological dynamics of a population carrying a loss-of-function variant at this locus.

Continued genetic surveillance of parasite populations is key to rapidly identify and respond to the emergence of treatment failure. In the recent emergence of artemisinin resistance in *Plasmodium falciparum*, genomic data has led to the identification of the major locus underlying resistance (*Ariey et al., 2014*; *Cheeseman et al., 2012*), revealed the genetic architecture of resistance (*Miotto et al., 2015*) and shed light on the population genetic context in which resistance is appearing (*Miotto et al., 2013*). Genomic surveillance is playing a key role in defining the geographic boundaries of the spreading artemisinin-resistant population. Failure of anti-*Leishmania* chemotherapy could become a similar public health emergency: miltefosine has shown reduced efficacy in both India (*Sundar et al., 2012*) and Nepal (*Rijal et al., 2013*). While amphotericin B is now being used against visceral leishmaniasis in ISC, few alternative treatments are available, and continued genomic surveillance will facilitate tracking the response of the *Leishmania* population to continued use of these drugs.

Monitoring drug resistance in clinical settings is challenging: the data set we present was generated as part of a five-year collaboration between clinicians in the endemic countries, parasitologists and genome biologists. This collaboration is critical in generating data that reflects the evolution of parasite populations in close to real time and as such is directly applicable in a public health context. The data we present here provide baseline information on the diversity of *Leishmania donovani* in the ISC that will contribute to future studies of drug resistance and epidemiology of this population. Our results show the promise of genomic surveillance for other *Leishmania* populations, where patient symptoms, the parasites involved and the main treatment modalities all differ from those in the ISC (*Sundar and Chakravarty, 2015*).

## Materials and methods

### Sample collection

The ethics committee of (i) the Nepal Health Research Council, Kathmandu, (ii) the Institute of Medical Sciences, Banaras Hindu University (BHU), Varanasi, India and (iii) the corresponding bodies at the Institute of Tropical Medicine of Antwerp and the Antwerp University, Belgium, reviewed and approved the study protocol. Informed written consent was obtained from each patient or his/her guardian for those <18 years of age. All the patients and caretakers/parents had the study purpose explained to them in local language.

A total of 204 parasite isolates were obtained from clinically confirmed VL patients in the high endemic regions of the Indian subcontinent (ISC) by the B.P. Koirala (BPK) Institute of Health Sciences in Dharan (Nepal, Terai, N=98), the Kala-azar Medical Research Center in Muzaffarpur (India, Bihar, N=98) and the Mymensingh Medical College in Mymensingh (Bangladesh, BD, N=8). The Indian and Nepalese isolates were collected as part of a multi-center collaborative project to investigate drug resistance in ISC and were all typed as *Leishmania (Leishmania) donovani*. Complete clinical and epidemiological data were available for the Indian and Nepalese isolates (*Supplementary file 1*).

The 204 *L. donovani* isolates were obtained from confirmed visceral leishmaniasis patients in previous clinical studies as described elsewhere (*Rijal et al., 2013*; *2007*). PCR-RFLP of the cysteine proteinase gene (*Quispe Tintaya et al., 2004*) typed all isolates as *Leishmania donovani*. Strain names consisted of 2–3 letters that indicated the location of isolation (BD, BHU, BPK), 2–4 digits that indicated the patient number in that location, a forward slash followed by 1–2 digits that indicated when the sample was isolated (0: before treatment, 1: 1 month after treatment, etc) and optionally the number of the parasite clone if the strain was cloned (clone one is listed as 'cl1'; clone two is listed as 'cl2', etc). Cloning was performed using the micro-drop method (*Van Meirvenne et al., 1975*). Patient treatment outcome was monitored at the end of treatment and at 3, 6 and 12 months post-treatment). Treatment non-response was defined as a case with positive parasitology at the end of the treatment course. Patients who were successfully cured at the end of treatment but in whom symptoms re-emerged within the 12 month follow-up period were classified as relapse cases. Patients who were cured at the end of treatment and remained cured within the 12 month follow-up period were classified as definite cures. If patients were lost to follow-up, the last known treatment

outcome was recorded. Seven pre- and post-treatment samples coming from the same patients were obtained. Patient treatment outcome after treatment with miltefosine (MIL) and pentavalent antimonials (SSG) was monitored during 12 months (at the end of treatment, month 3, month 6 and month 12 after treatment).

## Sample phenotyping

50 strains were phenotyped for their susceptibility to SSG using a standardized in vitro susceptibility test as described elsewhere (*Downing et al., 2011*; *Rijal et al., 2007*). An SSG-susceptible reference strain (BPK206/0cl10) was included in each assay. The classification into resistance and susceptible strains was determined by calculating the activity index (AI): the ratio of the EC50 of the strain in question versus the EC50 of the susceptible reference strain. AI values clustered strongly, with most strains showing an AI≤1 (25; classified as SSG-sensitive) or ≥6 (18; classified as SSG- resistant). A few strains (7) showed AI values around 3 and were considered as showing intermediate resistance.

## Genome sequencing

DNA isolation, sample preparation, DNA quantification and DNA library preparation were done as outlined previously (*Downing et al., 2011*). 100 bp paired-end sequence reads were generated (median coverage 44 per sample) with the Illumina Hiseq 2000 platform according to standard protocols. Read data are available under study ERP000140 at the European Nucleotide Archive (http://www.ebi.ac.uk/ena/data/view/ERP000140).

## DNA read mapping

Reads were mapped to the reference *L. donovani* genome BPK282/0cl4 using Smalt v5.7 (www.sanger.ac.uk/resources/software/smalt/). Options for exhaustive searching for alignments and random assignment of repetitively mapped reads were used to properly estimate read coverage. Non-mapping read exclusion, read file merging, sorting and elimination of PCR duplicates were implemented with Samtools v0.1.18 and Picard v1.85.

## Reference genome masking

The reference genome was masked at regions of the genome that were repetitive, duplicated, close to contig edges, structurally variable, or potentially mis-assembled. Five criteria masked a total of 6,358,203 bp out of the 32,444,998 bp reference genome sequence for *L. donovani* BPK282/0cl4, resulting in SNPs being called at 26,086,795 or 80.4% of the nuclear genome. Criteria were: 1. Manually identified repeats, commonly duplicated or deleted regions, regions with excessive rates of common SNPs and non-unique regions (*Downing et al., 2011*) identified with Gnuplot, the Artemis Comparison Tool, Artemis and Samtools tview (1,740,084 bp). 2. Duplicated regions determined by DNA similarity as Blast v2.2.25 (*Altschul et al., 1990*) hits between the two reference genome sequences for *L. donovani* BPK282/0cl4 and *L. infantum* JPCM5, with E-value less than 10e$^{-20}$ (2,082,546 bp). 3. Low complexity repeat regions determined by Tantan v0.13 (www.cbrc.jp/tantan/); (2,495,070 bp). 4. 100 bp regions adjacent to each contig edge (1,641,511 bp) – initially 13.8% of candidate SNPs were in these regions. 5. The first 300 bp and last 5 kb of all chromosomes, which are more likely to contain mis-assemblies.

## SNP detection using COCALL

SNPs were ascertained using a consensus calling approach (COCALL) that is based on the framework outlined for the 1000 Genomes project (*1000 Genomes Project Consortium, 2012*). COCALL applied five different variant detection approaches and combines evidence from them to calculate the support for each genotype. For complete details on the algorithm testing and development, see Appendix 1. In short, this approach avoids bias associated with systematic errors unique to each individual SNP caller by examining their consistency and identifying discordant mutations symptomatic of false positives. The five callers used were FreeBayes v0.9.5, GATK 2.0–38, Samtools Pileup v0.1.16 and Mpileup v0.1.18 based on the DNA read mapping by Smalt, and Cortex v1.0.5.13 based on its own *de novo* assembly and mapping. In a large population of genetically homogenous strains, superior inference power was achieved by examining the population-wide genotype at each candidate SNP site (i.e. population-based COCALL; *Figure 2—figure supplement 1*). Candidate SNPs

with genotype qualities of 40+ across all five callers were retained. SNPs with population normalized read depth ≤0.5 or ≥1.75 or with multiple derived alleles across the five callers were excluded. Candidate SNPs in soft-masked regions were accepted where the number of callers ≥3.5; those in non-masked regions were kept where the number of callers ≥2.5. SNP sites retained in the final set of retained SNP sites were supported by a mean of 4.5 callers out of 5.

## Copy number variant, somy and indel detection

Chromosomal read depths were computed using a trimmed median read depth (calculated as the median of read depths for sites with depths within one standard deviation of an initial, untrimmed, median read depth of each chromosome) and normalized as the depth per haploid genome as outlined previously (*Downing et al., 2011*). Somy levels were estimated as the median normalized chromosomal read depths (*Downing et al., 2011*). Local copy number variants (CNVs) were detected where the local read depth was significantly different from the median depth of approximately 60 samples from ICS4, ISC6 and ISC8 whose depth profile is similar to that of BPK282/0cl4, and were measured with respect to the haploid depth to exclude somy variability. Two CNVs in particular, the MAPK1 and H-locus, were further investigated as they show functions potentially relevant to parasite adaptation (*Downing et al., 2011*). A quantitative PCR assay in a subset of 46 samples was performed to confirm the copy number variation of the MAPK1 and H-locus amplicons. The nature of the amplification (extra- or intra-chromosomal) was determined by pulsed-field gel electrophoresis (PFGE) and southern blot hybridization comparing two strains that showed differential amplification of these loci (ISC6 strain BPK282/0cl4: amplification; and ISC1 strain BPK026/0cl5: no amplification). To exclude the possibility that the amplicons are a culturing artefact, PCRs using primers that enabled amplification of circular episomes or tandem duplications was also attempted directly on five bone marrow samples from VL patients. Indels were detected using a consensus calling method based on the concordance of results across four tools: Cortex, Freebayes, GATK and Samtools Mpileup. For complete details on the Somy, CNV, indel and episome detection, see Appendix 2.

## Haplotype inference and linkage disequilibrium

Haplotypes were inferred using PHASE v2.1.1 (*Stephens et al., 2001*): 0.1% of genotypes in the Core 191 and 0.9% in ISC1 had confidence scores <0.95. Haplotypes were inferred with a general recombination rate model (*Li and Stephens, 2003*) with ten runs, each with a burn-in of 100 steps, 100 iterations and a single MC thinning step and recombination rate estimated for each chromosome. Convergence was examined for each chromosome: recombination estimates were consistent, though there was more variation between phasing runs for chromosome 16 in the core population and consequently inferred haplotypes are less certain for that chromosome. There was no correlation between the mean chromosome copy number and mean recombination rate or PHASE probability values for inferred haplotypes ($r^2$=0.011). While variation in somy is not explicitly accounted for in the phasing process, the rapid flux in the somy levels of aneuploid chromosomes may mean this variation has no effect on haplotype inference. Of 3,567 heterozygous sites, 3,076 (86%) had a PHASE probability of exactly 100% and 437 had PHASE probabilities < 0.95: these lower-confidence haplotypes were masked. Haplotypes for BHU1087/0 were inferred along with the core population. The phased core population SNP set had 2,401 SNPs: 17 singletons were masked. The smaller sample size meant that phasing within ISC1 was less successful: phase was successfully inferred for 2,308 sites using the same (0.95) confidence score threshold: 524 were not phased and were excluded from further analysis. No correlation between phasing confidence score and trisomy or tetrasomy was apparent.

Linkage disequilibrium (LD) was inferred as the correlation in genotypes ($r^2$ values) between SNP pairs using Bcftools v0.1.17 screened with Samtools Mpileup given SNP mapping qualities >30 and base qualities >25. These pairwise $r^2$ values were used to examine genome-wide LD patterns and LD decay with distance. Recombination was confirmed using the four-gamete test (*Hudson and Kaplan, 1985*). Mean chromosomal estimates of LD in the core population did not correlate with somy level if the tetrasomic chromosome 31 was excluded ($r^2$=0.001) but did if chromosome 31 was included ($r^2$=0.167). Somy had little impact on the variance of LD per chromosome ($r^2$=0.017 with chr31, $r^2$=0.000 without chr31). Variance in somy level across chromosomes had no association with either the mean or variance of LD per chromosome. We calculated zygosity as the probability that a SNP

exists at a distance $d$ from a SNP at a site $x$ assuming diploidy (**Lynch, 2008**). No differences between homozygous and heterozygous SNP clustering measured as a product of chromosomal distance was observed.

## Population genomic identification of groups

*L. infantum* JPCM5 (MCAN/ES/1998/LLM-877) from Spain and LV9 (MHOM/ET/1967/HU3) from Sudan were used (**Downing et al., 2012**) for comparison with the *L. donovani* genomes generated in this study. Variants were called for these two samples using the approaches outlined above. For two additional *L. donovani* isolates from Sri Lanka (**Zhang et al., 2014**), we mapped Illumina GAII reads using Smalt v5.7 as above and called candidate SNPs at non-masked regions using Samtools Pileup v0.1.16 (**Li et al., 2009**), followed by screening steps as above. Two genomes were excluded in the final analyses because sequence reads were of insufficient quality (for MHOM/IN/10/ BHU1087/0) or because of a suspected mixed infection (for MHOM/IN/10/BHU790/0). BHU790/0 is distantly related to the core population (most likely ISC3) and appears to be a mixed infection rather than a hybrid because its average read allele frequency of heterozygous SNPs approximates 0.17, whereas most detected hybrids had mean read allele frequencies of 0.4–0.5. Remaining data were used to construct phylogenies using the 211,536 sites containing verified SNPs in the entire sample set (ISC1/2/3/4/5/6/7/8/9/10 and ungrouped, LV9, JPCM5, BPK512/0cl9). JPCM5, LV9, the two Sri Lankan isolates and one sample from our collection (BPK512/0cl9) represented genetically distinct lineages, distinct to both the ISC1 (n=12) and core populations (n=191). Seven SNPs in the core population and ten in ISC1 had multiple derived alleles compared to reference genome sample BPK282/ 0cl4 (**Supplementary file 2, table t**). These were included in all diversity analyses, but not those involving phased haplotypes.

Genome-wide phylogenetic trees were constructed with RAxML v8.1.1 (**Stamatakis, 2014**) using the GTR+G substitution model and 1000 bootstrap replicates for 10 runs for the core population (881 alignment patterns), ISC1 (349 alignment patterns), and all samples including the CL and VL samples from Sri Lanka (**Zhang et al., 2014**) (2274 alignment patterns). The best fitting substitution model determined using MEGA v6 (**Tamura et al., 2011**) for the core population was GTR+G. The final phylogenies were visualised using MEGA v6 (**Tamura et al., 2011**) and Splitstree v4 (**Huson and Bryant, 2006**). Unrooted haplotype trees for the phased SNPs for each chromosome were constructed from maximum-likelihood distances for the TN93 substitution model using the package Ape (**Paradis et al., 2004**) v3.1–4 in R version 3.12.

Samples in the core population of 191 isolates were classified using model-based clustering as implemented in Structure v2.3.2.1 (**Pritchard et al., 2000**) and principal component analysis (PCA) of the allele frequencies. Given a number of genetically distinct clusters (K), samples were probabilistically assigned to a population independent of a mutation model with a prior of 1/K based on the correlation in genotypes of each sample with estimated population allele frequencies. $1 \leq K \leq 15$ was examined with admixture and incomplete membership allowed to reduce overfitting. We used $10^5$ burn-in steps before a run of $2 \times 10^6$ steps with three independent runs per K to confirm chain convergence. The most likely number of clusters was based on the second-order rate of change of the likelihood function ($\Delta$K, **Evanno et al., 2005**). At K=4 the groups were composed of ISC2/3/9/10, ISC4, ISC5 and ISC6/7/8. Inter-population differentiation was lower for ISC2/3/9/10 ($F_{ST}=0.36$) compared to the others ($0.85 < F_{ST} < 0.98$). K=7 was the most probable K value ($\Delta$K=25.8): the groups were composed of ISC2, ISC3, ISC4, ISC5, ISC6/7/8, and ISC9/10 (all $F_{ST} > 0.79$) – the 21 ungrouped samples collectively had an $F_{ST}=0$. Most population membership assignments were >0.97 with few ambiguous values (range 0.80–0.97). At K=9, ISC6/7/8 split into ISC6 and ISC7/8 (both $F_{ST} > 0.85$). At K=10, ISC7/8 segregated into ISC7 and ISC8.

## Inference of historical population sizes, geographic locations and migration rates

Dated phylogenies, historical population sizes and migration patterns were modelled for the 191 core clinical isolates using BEAST v1.8.1 (**Drummond et al., 2012**). For molecular clock analyses, hybrid isolates not assigned to any of the ISC groups were removed from the dataset, as were the Bangladeshi outgroups for most analyses. Dates for each were fixed to the month of isolation, with sampling dates for those for which only isolation year data was available estimated during the

MCMC but given a uniform prior on sampling ages within that year. Broadly consistent date estimates were obtained under three different models: with an uncorrelated lognormal relaxed clock model and a TVM substitution model and a Bayesian skyride model for population sizes, under the same model but with a strict clock model and finally under a GTR substitution model, with a simple constant population size coalescent model for data including the outgroups. Migration rate estimates were obtained by including a simple continuous-time Markov model of a discrete trait representing the country (Nepal/India) of isolation, so that ancestral states and rates of change in geographical location were estimated along the phylogeny. All analyses were made with a minimum of 8 independent MCMC runs, for 200 million update generations per run. Convergence was assessed by inspection in Tracer v1.6, confirming that at least 5 of the 8 runs had converged to the same stationary distribution of parameters and that this had the highest likelihood. In most analyses, seven or eight chains all converged to the same posterior distribution, but the Bayesian skyride analyses converged more slowly. ESS estimates for almost all parameters across runs was over 500, except for some skyride population size parameters. The first 20 million generations of each MCMC run were removed before combining all converged runs for inference. Historical population sizes were estimated both with the Bayesian skyride model and by transforming lineage-through-time data for all trees in the posterior probability distribution from the strict clock model above using the package Ape (*Paradis et al., 2004*) v3.1–4 in R version 3.12. To compare population sizes between the drug resistant clade and others, we split ISC5 from other data and removed coalescent events between the ISC groups (the oldest six) to make these comparable with the ISC5 coalescence.

## Population genomic identification of admixture using allele frequency correlations

f-statistics describe the correlation in allele frequencies between populations (*Patterson et al., 2012*; *Reich et al., 2009*). The simplest ($f_2$) is simply the sum-of-squares difference in allele frequency between two populations averaged across loci, and so captures the amount of divergence, or branch length between two populations. Two more complex statistics, $f_3$ and $f_4$ are calculated as differences between $f_2$ statistics between groups of 3 and 4 related populations. $f_3(C;A,B)$ has the property that, for a population C derived from populations A and B, it is expected to be positive if A,B and C are related by a simple history of divergence and genetic drift, but negative if admixture from A or B has contributed to the genetic composition of population C, while being robust to the details of the relationship between the three populations. In contrast, the value of $f_4(A,B,C,D)$ does depend on the evolutionary history of populations A, B, C and D and so can be used to test a proposed relationship: if the four populations are related as ((A,B),(C,D)) the $f_4$ statistic is expected to be zero; for ((A,C),(B,D)) it is expected to be positive and for ((A,D),(B,C)), negative. Finally, if the evolutionary history of three ancestral populations is known, the ratio of two $f_4$ ratios is an estimate of the relative contribution of two potential parental populations to a fourth admixed population, given an outgroup.

## Population genomic identification of admixture using haplotype sharing

Whereas groups ISC2/3/4/5/6/7 seemed clearly defined phylogenetically and by Structure, ISC8/9/10 were not and no simple relatedness among the 21 ungrouped samples was detected. Consequently, we used inferred haplotypes to test whether these represented genetically discrete populations, or whether some of those samples were mixtures of ISC3/4/5/6/7 generated by hybridisation between these groups (*Lawson et al., 2012*). Chromopainter v0.0.2 and FineStructure v0.0.2 inferred ancestral patterns of haplotype similarity among samples without a prior assumption of a given number of populations or of independence between mutations.

Co-ancestry matrices for the core population were computed using Chromopainter v0.0.2 as the number of segments potentially donated to or received from individual samples, using the phased haplotypes. Recombination rates between pairs of SNPs inferred by PHASE were used for each of 36 unlinked chromosomes. Groups of SNPs on a single chromosome were expected to be exchanged as blocks of different sizes, so a higher number and longer lengths of shared blocks indicate recent common ancestry. The most likely ancestral sample or population was assigned according to its similarity to corresponding segments in a set of donor isolates. Two main datasets were generated by ChromoPainter: a co-ancestry matrix where all 191 could donate to all 191 as

recipients (191x191), and another where six representative samples were used as the only donors (191x6: BD09 for ISC2, BPK067/0cl2 for ISC3, BPK087/0cl11 for ISC4, BPK275/0cl18 for ISC5, BPK282/0cl4 for ISC6, BHU200/0 for ISC7). The expected number of chunks was minimised for the six representative samples, with k=80 segments and an effective number of chunks c=0.02. Reducing the number of representative strains to represent distinct groups identified by Structure with smaller K parameters resulted in smaller k and larger c values, suggesting that using six representative samples was the optimal number for discrimination within the core population. Though ISC7 was a subset of ISC6, ISC7 had a large number of fixed SNPs sufficient to differentiate it from ISC6 with Structure, so it was included. For the 191x191 comparison, k=79 segments was expected and the effective number of chunks was lower (c=0.00054) because the total diversity of the donor set per SNP had decreased.

These 191x191 and 191x6 co-ancestry matrices represented the most probable number of segments copied from each donor to each recipient, and also the relative probability of ancestry across the set of donors for each SNP for each sample. The number of donors per recipient was set to 100. 20 expectation-maximisation algorithm iterations was sufficient to maximise the recombination-scaling coefficient ($N_e$) and copying probabilities with 10<k<1000 iterations across different number of donor samples assuming a minimum recombination rate of $10^{-15}$ Morgans/bp. For the 191x191 matrix, the $N_e$=523.3 and the mutation rate ($\mu$) was 0.000181. For the 191x6 matrix, the $N_e$=1015.1 and $\mu$=0.000628: $N_e$ and $\mu$ were higher because there were more mutations per sample.

The 191x191 matrix was clustered for $10^6$ MCMC (Markov chain Monte Carlo) steps with a burn-in of 10,000 and a skip of 100 steps using FineStructure v0.02 to obtain aggregated expected segment sharing between samples and populations with 100 trees examined per merge step. This distinguished complex ancestral patterns of segment sharing for the strains which Structure could not fully assign to single populations.

To verify FineStructure and Structure results, the correlation in the SNP allele frequencies across samples was examined in the core population for six principal components with $p<10^{-7}$ using PCA implemented by smartPCA in Eigensoft v4.2 (*Price et al., 2006*). The first PC separated ISC2 (10.1% of all variation), the second ISC4 (6.4%), the third ISC5 (5.8%), the fourth ISC3 (4.9%), the fifth BPK035/0cl1 and BPK043/0cl2 (4.2%) and the sixth a subset from ISC9/10 (3.9%). This was repeated for the 2353 variable sites in the core population (ISC3/4/5/6/7/8/9/10 and ungrouped samples, n=183) excluding the 8 samples from Bangladesh (ISC2). This partitioned ISC5 (PC1, 7.4%), then ISC4 (6.8%), third ISC3 (5.6%), and fourth BPK035/0cl1-BPK043/0cl2 (4.8%). Eigenstrat and FineStructure PCA results were effectively the same but with some different axis labels – PC1 in the former was PC3 in the latter. FineStructure 191x191 ancestry patterns partitioned ISC4 vs ISC6 over PC1 (16.8% of variation), and ISC5 vs ISC6 over PC2 (15.5%). The next (12.3%) differentiated ISC2, and PCs 4 (6.4%) and 5 (5.7%) separated ISC3. PC6 in FineStructure differentiated the BPK035/0cl1-BPK043/0cl2 pair.

## Population genomic identification of drug-resistance elements

Information on in vitro $Sb^V$-resistance was available for 50/191 Core 191 isolates, from which 25 were sensitive and 25 resistant (*Supplementary file 1*). Links between genetic diversity (SNP, indel, CNV and somy) and in vitro $Sb^V$-resistance were assessed using the Fisher Exact test (FET), Mann Whitney U-tests (MWU) and odds ratios (ORs), implemented on 103 CNVs and 17 indels (in 14 genes) as well as 2,392 phased SNPs genotypes. SNPs were assigned to the 5' and 3' UTR if they were within 1 kb of the start or end of the gene (respectively). To counter bias associated with the small sample size, FET and MWU were used initially. For the FET, variants were defined as discrete variables: SNPs as 0, 1 or 2 non-reference alleles, and small indels as the diploid number of inserted or deleted basepairs. For the MWU, mutations were considered as a continuous variable such that the somy state was the haploid chromosome state, and CNVs were the haploid copy number times the somy state. The null hypothesis was that there were no significant genetic differences between $Sb^V$-R and $Sb^V$-S strains (subject to $p<0.01$). The FETs and MWU were limited by the partial association of different mutations with the phenotypes, so we examined ORs of the derived alleles segregating in multiple ISC populations with 6+ non-reference alleles for which the absolute difference in $Sb^V$-R and $Sb^V$-S allele frequencies >0.1 using the log-scaled EC50 values. We compared the log-scaled EC50 values of each allele pair using t-tests.

We also examined samples for which the patient was treated with Sb$^V$ and was either cured or not, samples for which the patient was treated with miltefosine (MIL) and was either cured or not, and also in vitro MIL resistance levels as implemented above for Sb$^V$.

## Testing for selective processes among ISC populations

Evidence of historical differences in selective processes on the ancestors of the major ISC populations was assessed as the rate of accumulation of derived alleles. Stronger purifying selection should purge deleterious derived alleles more quickly, detected as an excess of nonsynonymous changes relative to synonymous ones, as previously observed for ISC isolates (*Downing et al., 2011*). This signature should be most apparent for derived alleles, which should accumulate at a net rate dependent on the historical effective population sizes and selective coefficients. Using *L. infantum* JPCM5 as the outgroup, the relative abundance of derived alleles in one population that were absent in the other for each ISC population pair (ISC2-7) were determined as the statistic R (*Do et al., 2015*). The associated ratio R2 denoted the relative rate of homozygous derived allele accumulation between populations. R and R2 should approximate 1 assuming no difference in the strength of selection, and primarily depend on the derived allele frequency per population, so the main confounder was variance in historical effective population sizes among ISC populations. To calculate confidence intervals for these R values that take into account correlation between neighbouring sites, we used a Weighted Block Jackknife by splitting the SNPs according to chromosome (*Busing et al., 1999*) to counter the extensive linkage disequilibrium between SNPs (*Moorjani et al., 2011*): discrete chromosomal blocks may still be linked. This was adjusted for the number of SNPs per block to reflect the variability in the relative selective pressure (*Kunsch, 1989*). A threshold of four times the standard error of these jackknife estimates was used as a criteria for identifying comparisons deviating significantly from expected values (*Do et al., 2015*).

## AQP1 modelling

A protein model of the intact *Leishmania donovani* AQP1 from BPK282/0cl4 was created using MODELLER 9.14 (*Sali and Blundell, 1993*). The template for homology modelling was the crystal structure of the aquaglyceroporin from *Plasmodium falciparum* in complex with glycerol (PDB code: 3c02) published by Newby and co-workers (*Newby et al., 2008*). The sequence identity between the target and the template was approximately 33%. PyMOL version 1.50.04 (Schrödinger) was used to generate the biological units for the aquaglyceroporin from *Plasmodium falciparum* (generation of symmetry mates function in pymol). The C-alpha atoms of chain A, B, C and D of the tetramer template were restrained during homology modeling using MODELLER in order to reduce the number of interatomic distances that needed to be calculated.

## Acknowledgements

The authors gratefully acknowledge financial support for the Kaladrug-R consortium from the European Union framework program (FP7-222895). JCD, BO, MBo are supported by the Belgian Development Cooperation (FA3 II VL control and FA3 project 95502), HI, FVdB, AM, MV, MBe, GD, FD, IM, and JCD by the Belgian Science Policy Office (TRIT, P7/41) and the Flemish Fund for Scientific Research (G.0.B81.12) and MBe by the INBEV-Baillet Latour foundation. JAC, MB, MJS are supported by the Wellcome Trust through their core funding of the Wellcome Trust Sanger Institute (grant 098051) and thank members of the DNA pipelines team at WTSI for generating the sequencing libraries.

# Additional information

## Funding

| Funder | Grant reference number | Author |
| --- | --- | --- |
| European Commission | EU framework program FP7-222895 | Hideo Imamura<br>Tim Downing<br>Frederik Van den Broeck<br>Mandy J Sanders<br>Suman Rijal<br>Shyam Sundar<br>An Mannaert<br>Manu Vanaerschot<br>Maya Berg<br>Géraldine De Muylder<br>Franck Dumetz<br>Ilse Maes<br>Saskia Decuypere<br>Keshav Rai<br>Surendra Uranw<br>Narayan Raj Bhattarai<br>Basudha Khanal<br>Vijay Kumar Prajapati<br>Smriti Sharma<br>Olivia Stark<br>Gabriele Schönian<br>Syamal Roy<br>Bart Ostyn<br>Marleen Boelaert<br>Matthew Berriman<br>Jean-Claude Dujardin<br>James A Cotton |
| Belgian Science Policy Office | TRIT, P7/41 | Hideo Imamura<br>Frederik Van den Broeck<br>An Mannaert<br>Manu Vanaerschot<br>Maya Berg<br>Géraldine De Muylder<br>Franck Dumetz<br>Ilse Maes<br>Benoit Vanhollebeke<br>Jean-Claude Dujardin |
| Flemish Fund for Scientific Research | G.0.B81.12 | Hideo Imamura<br>Frederik Van den Broeck<br>An Mannaert<br>Manu Vanaerschot<br>Maya Berg<br>Géraldine De Muylder<br>Franck Dumetz<br>Ilse Maes<br>Jean-Claude Dujardin |
| Department of Economy, Science and Innovation in Flanders | ITM-SOFIB | Hideo Imamura<br>Manu Vanaerschot<br>Maya Berg<br>Saskia Decuypere<br>Jean-Claude Dujardin<br>Malgorzata Domagalska |
| Wellcome Trust | 098051 | Mandy J Sanders<br>Matthew Berriman<br>James A Cotton |
| Belgian Development Cooperation | FA3 II VL control and FA3 project 95502 | Suman Rijal<br>Keshav Rai<br>Surendra Uranw<br>Narayan Raj Bhattarai<br>Basudha Khanal<br>Bart Ostyn<br>Marleen Boelaert<br>Jean-Claude Dujardin |

| INBEV-Baillet Latour Foundation | | Maya Berg |
| --- | --- | --- |
| Flemish Fund for Scientific Research | 11O1614N | Bart Cuypers |
| JC Bose National Fellowship, DST, Government of India | SB/S2/JCB-65/2014 | Syamal Roy |
| Council of Scientific and Industrial Research | BSC0114 | Syamal Roy |

The funders had no role in study design, data collection and interpretation, or the decision to submit the work for publication.

## Author contributions

HI, Developed and verified SNP and indel detection scheme COCALL, Aligned reads and determined read depth and performed genetic variant detection and correlation analysis among genetic variants, Analysis and interpretation of data, Drafting or revising the article; TD, Developed and verified SNP and indel detection scheme COCALL, Analysed population structure and performed phylogenetic analyses, Examined the correspondence of mutation with drug phenotype variation, Analysis and interpretation of data, Drafting or revising the article; FVdB, Analysed correlations of aneuploidy with heterozygosity, Examined the correspondence of mutation with drug phenotype variation, Performed additional sequence data analyses, Analysis and interpretation of data, Drafting or revising the article; MJS, Coordinated sequence materials and sequencing. Reviewed the final manuscript., Acquisition of data; SRi, SSu, SU, Coordinated sample collection and provided and documented samples and clinical data, Reviewed the final manuscript, Acquisition of data, Contributed unpublished essential data or reagents; AM, Performed molecular-clock and phylogeographic analyses and assessed epidemiological and phylogenetic history, Performed additional sequence data analysis and additional experimental analysis and verification, Reviewed the final manuscript., Analysis and interpretation of data; MV, Coordinated sample collection and provided and documented samples and clinical data, Coordinated sequence materials and sequencing, Performed additional sequence data analysis, Acquisition of data, Analysis and interpretation of data, Drafting or revising the article, Contributed unpublished essential data or reagents; MBe, Performed additional sequence data analysis, Reviewed the final manuscript, Analysis and interpretation of data; GDM, Performed amplicon verification, Performed AQP1 modelling and analysis, Performed additional experimental analysis and verification, Reviewed the final manuscript., Acquisition of data, Analysis and interpretation of data; FD, BC, Performed additional experimental analysis and verification, Reviewed the final manuscript, Acquisition of data; IM, Conducted drug susceptibility assays, Coordinated sequence materials and sequencing, Performed additional experimental analysis and verification, Reviewed the final manuscript., Acquisition of data, Contributed unpublished essential data or reagents; MD, Performed analyses on aneuploidy. Reviewed the final manuscript; SD, Co-ordinated sample collection and provided and documented samples and clinical data, Co-ordinated sequence materials and sequencing. Reviewed the final manuscript, Acquisition of data, Contributed unpublished essential data or reagents; KR, Conducted drug susceptibility assays, Performed additional sequence data analysis, Performed additional experimental analysis and verification, Reviewed the final manuscript, Acquisition of data, Analysis and interpretation of data; NRB, LM, Conducted drug susceptibility assays, Reviewed the final manuscript, Acquisition of data, Contributed unpublished essential data or reagents; BK, OS, GS, BO, MBo, Co-ordinated sample collection and provided and documented samples and clinical data, Reviewed the final manuscript, Acquisition of data, Contributed unpublished essential data or reagents; VKP, Coordinated sample collection and provided and documented samples and clinical data, Conducted drug susceptibility assays, Reviewed the final manuscript, Acquisition of data, Contributed unpublished essential data or reagents; SSh, Co-ordinated sample collection and provided and documented samples and clinical data, reviewed the final manuscript, acquired data, contributed unpublished essential data or reagents; HPDK, LS, Performed AQP1 modelling and analysis, Reviewed the final manuscript, Acquisition of data, Analysis and interpretation of data; BV, Performed amplicon verification, Performed additional experimental analysis and verification, Reviewed the final manuscript, Acquisition of data, Analysis and interpretation of data; SRo, Co-ordinated sample collection and provided and documented samples and

clinical data, Conducted drug susceptibility assays, Performed additional experimental analysis and verification, Reviewed the final manuscript, Acquisition of data, Analysis and interpretation of data, Contributed unpublished essential data or reagents; MB, Conception and design, Drafting or revising the article; J-CD, Co-ordinated sample collection and provided and documented samples and clinical data, Performed molecular-clock and phylogeographic analyses and assessed epidemiological and phylogenetic history, Performed additional sequence data analysis, Conception and design, Acquisition of data, Analysis and interpretation of data, Drafting or revising the article, Contributed unpublished essential data or reagents; JAC, Analysed population structure and performed phylogenetic analyses, Performed molecular-clock and phylogeographic analyses and assessed epidemiological and phylogenetic history, Performed additional sequence data analysis, Analysis and interpretation of data, Drafting or revising the article

## Author ORCIDs
Tim Downing, http://orcid.org/0000-0002-8385-6730
Frederik Van den Broeck, http://orcid.org/0000-0003-2542-5585
Keshav Rai, http://orcid.org/0000-0002-9747-3431
Harry P De Koning, http://orcid.org/0000-0002-9963-1827
Benoit Vanhollebeke, http://orcid.org/0000-0002-0353-365X
Marleen Boelaert, http://orcid.org/0000-0001-8051-6776
Matthew Berriman, http://orcid.org/0000-0002-9581-0377
James A Cotton, http://orcid.org/0000-0001-5475-3583

## Additional files

### Supplementary files

• Supplementary file 1. DNA samples and their origin. DNA samples are ordered according to their classification into genetic populations as described in the materials and methods section. The number after the last forward slash in the isolate name indicates when the sample was isolated from the patient in terms of months after onset of a one-month treatment. Patients were followed up for 12 months to assess treatment outcome, with a few exceptions that were lost to follow up and for which the last known outcome is shown mentioning the last recorded time point. The SSG activity index is calculated by dividing the IC50 of a specific strain by the IC50 of BPK206/0cl10 (a sensitive control included in all SSG-susceptibility tests). Activity indexes above 3 are considered to indicate SSG-resistance. MIL: Miltefosine, SSG: sodium stibogluconate, P: passage number, x: unknown number of passages between isolation and shipment to research institute for DNA preparation.

• Supplementary file 2. (a) Numbers and types of mutations in the core population compared to genetically distinct strains LV9 (MHOM/ET/1967/HU3) and JPCM5 (MCAN/ES/1998/LLM-877), as well as the ISC1/2. N/S ratio denotes the number of nonsynonymous SNPs divided by the synonymous ones. The 'LV9+JPCM5-191 lineage' refers to homozygous SNPs occurring in the core population ancestral branch only since its divergence with LV9 and JPCM5. The 'ISC1-191 lineage' refers to fixed homozygous SNPs between all of the core population and all of ISC1. The 'ISC2-183 lineage' refers to fixed homozygous SNPs between ISC2 (Bangladesh) and ISC3/4/5/6/7/8/9/10 and ungrouped isolates (India or Nepal). (b) Number of SNPs unique to each population or lineage. SNPs were determined as unique where their intraspecific frequency was >0.95 and <0.05 in all other samples (excluding ungrouped samples). The number of samples did not reflect the number of SNPs or haplotypes: haplotype diversity was >0.99 for all groups except ISC7 (0.34). (c) Genes ordered by ID with two or more SNPs and π/kb>1. The non-synonymous variant V185M in a NADH-flavin oxidoreductase/NADH oxidase gene (LdBPK_120730) was present here in the ISC1/2/3 as well as previously being observed in Kenyan *L. donovani* (*Downing et al., 2012*). 956 genes had at least one SNP, of which 566 had SNPs that were not singletons. Only 63 genes contain ≥2 SNPs and 12 genes contain ≥3 SNPs. (d) Mean numbers of SNP differences within and between groups. Rows and columns denote comparisons: within populations, the mean number of SNPs between strain pairs (π) is shown; and between populations, the mean number of SNPs between samples from each population is given. Within the core population, there was an excess of singletons (Tajima's D=-2.2,

p<0.01, Fu and Li's D=-0.78, Fu and Li's F=-1.87: the values were the same with either JPCM5 or LV9 as the outgroup). 62.6% of SNPs occurred in one sample only and the correlation of the number of homozygous and heterozygous SNPs per sample was small ($r^2$=0.005). 291 distinct haplotypes out of a maximum possible of 382 were resolved in the phased SNPs, with a resulting haplotype diversity (Hd) of 0.994. Nepalese samples were on average more diverse compared to the Indian ones ($\pi$=84.3 per Mb vs 66.5, Hd=0.996 vs 0.982). ISC6 was restricted to Nepal ($\pi$=12.2 per Mb). (e) Branch distances between groups using the 2 population (f2) statistic. The scaled correlation in allele frequencies were computed for each reference group (top row) and groups (first column). ISC9 and ISC10 were examined together as one group. The distribution of allele sharing across ISC3/4/5/6 in subsets of trios was compared to that for ISC2 using F-statistics. The relative pairwise correlation in allele frequencies between populations using the f2 test indicated ISC2 as the most consistently divergent population and also highlighted distinctive allele frequencies in the recently emerged Indian ISC7 group. In contrast, groups with possible admixture (ISC9 and ISC10) had much shorter branch lengths. (f) f4 ratios for candidate admixed populations for mixture from ISC5, ISC6 and ISC7.Each ratio was computed as $\frac{f_4(A,outgroup;B,X)}{f_4(A,outgroup;B,ISC5)}$ with A and B as either ISC6 or ISC7. Populations X were the 6 samples identified as possible mixtures of the ancestors of ISC5 and ISC6/7: BHU815/0, BHU764A1, BHU274/0, BHU574cl4, BHU581cl2, and BHU572cl3. The gene flow from ISC5 was the f4 ratio and that from B (ISC6 or ISC7) was 1-f4. As expected, if ISC6 and 7 grouped together relative to other groups (f4>0). f4 statistics were considered to support a particular relationship between four individual isolates if 0.05<f4<0.05. ISC2 clustered with ISC3 much more frequently (66.7%) than any other (ISC4, ISC5, ISC6 all 22.2% each). ISC3 had drift patterns more similar to ISC4 (44.4%) than ISC6 (22.2%) or ISC5 (0%). ISC4 had greater correlation with ISC5 (44.4%) than ISC6 (22.2%). ISC5 and ISC6 had more similar drift in 66.7% of tests. These values are consistent with the relationship (ISC2,(ISC3,(ISC4,(ISC5,ISC6))))) shown by the phylogenetic analysis. Admixture levels were measured for the six ISC5-ISC6/7 hybrids as a single population using f4 ratios with ISC2, ISC3 and ISC4 as outgroups. The six hybrids were BHU815/0, BHU764A1, BHU274/0, BHU574cl4, BHU581cl2 and BHU572cl3. The fraction of ancestry attributed to ISC5 was 54–56% and to ISC6 was 44–46% for ISC5-ISC6 mixing, and the alternative ISC5-ISC7 mixture had ISC5 at 59–61% and ISC7 at 39–41%. (g) Diagnostic SNPs for ISC7. 20 mutations were unique to ISC7 compared to the other genetically homogeneous populations (ISC2/3/4/6/7/8/9/10) with a ISC7 frequency >0.98 and a non-ISC7 frequency <0.02. Six were nonsynonymous. These SNPs coincided with a distinct chromosome copy number pattern in ISC7. SNPs were assigned to the 5' and 3' UTR if they are within 1 Kb of the start or end of the gene (respectively). ISC7 was a recent radiation restricted to India ($\pi$=1.8 per Mb). (h) Diagnostic SNPs for ISC5. 32 mutations were unique to ISC5 compared to the other genetically homogeneous populations (ISC2/3/4/6/7/8/9/10), with ISC5 frequency >0.95 and a non-ISC5 frequency <0.05. Eight changes were nonsynonymous. *Nine were previously associated with antimonial resistance, including four samples from this ISC5 dataset (*Downing et al., 2011*): six of these would change the amino acid sequence. SNPs were assigned to the 5' and 3' UTR if they are within 1 Kb of the start or end of the gene (respectively). (i) Diagnostic SNPs for disomic clade within ISC5. 16 diagnostic SNPs unique to a subset of the ISC5 associated with an expansion of disomic strains from polysomic ancestors (BHU1216/0, BHU239/0, BHU271/0, BHU267/0, BHU800/1, BBU269/0, and BHU273/0). Each SNP had a read-depth allele frequency >0.978 within this group and a frequency <0.04 in the other strains not in this group. These SNPs coincided with a distinct chromosome copy number pattern in this group. Ref is the reference genome BPK282/0cl4 allele and Var is the observed derived one. (j) Linkage disequilibrium (LD) and SNP density across chromosomes in the core population. LD was computed as the $r^2$ values between SNP pairs. SD is the standard deviation. The number of homozygous (hom) and heterozygous (het) mutations per chromosome across all 191 samples is shown. LD was markedly higher for chromosome 31, and lower for 1, 6, 7, 8 and 10. The number of recombinant pairs was largest on chr31 (469), which had the highest somy level and was the most SNP-dense chromosome. (k) A genomic map of SNPs in the core population whose ancestry could be assigned to a population. Using the 191x6 ChromoPainter model, the 610 ancestry-informative SNPs were examined to highlight those with probabilities of being donated given ISC group was >0.4: ISC2 (gold), ISC3 (grey), ISC4 (red), ISC5 (green), ISC6 (blue) and ISC7 (light blue). The representative samples used for the 191x6 ChromoPainter model were BD09 (ISC2), NEP123/6 (ISC3), BPK087/0cl1 (ISC4), BPK275/0cl18 (ISC5), BPK282/0cl4

(ISC6) and BHU200/0 (ISC7). SNPs with probabilities <0.4 are uncoloured. For chr32 and chr35, most recombinant pairs could be attributed to the eight ungrouped hybrid isolates. (l) Number and heterozygosity of ancestry-informative SNPs per isolate. The number of homozygous (hom) and heterozygous (het) SNPs distinguishing each core sample (191 total) from six representative samples from each genetically distinct ISC population: ISC2 (BD09), ISC3 (NEP123/6), ISC4 (BPK087/0cl1), ISC5 (BPK275/0cl18), ISC6 (BPK282/0cl4) and ISC7 (BHU200/0). There were 610 informative SNPs with an ancestry probability threshold >0.4. The relative ratio of homozygous and heterozygous SNPs denoted the genetic relatedness of the strains. Samples possessing recent common ancestry with the representative strains had few homozygous SNPs but the same rate of heterozygous SNPs compared to other representative samples. Similarly, samples possessing recent common ancestry with multiple representative samples had fewer homozygous SNPs and no change in the heterozygous SNPs. This was apparent for highlighted ungrouped samples BHU815/0, BHU764A1, BHU274/0, BHU574cl4, BHU581cl2, BHU572cl3 (all ISC5/6/7: green and blue); and BHU744/0, BHU774/0 (ISC6/7: blue). In many groups haplotype segments were shared across isolates widely separated in sampling date. For example, in ISC3, 35 haplotype segments were shared between NEP123/6 (2004), BPK507/0 (December 2009), BPK515/0, BPK518/0 and BPK519/0 (all February 2010); 53 haplotype segments were shared between BPK157/0cl5 (October 2002) and BPK602/0 (February 2011) (both ISC9); 44 between BPK280/0 (August 2003) and BPK615/0 (March 2011; both ungrouped); 31 between BHU274/0 (ungrouped, August 2007) with BHU1199/0 (ISC9, November 2010). (m) Distribution of copying probabilities of SNPs in two representative isolates. Expected copying probabilities from Chromopainter assigning ancestry to SNPs from BHU581/0 (Ungrouped, left) and BPK275/0cl18 (ISC5, right) using probabilities derived from a model assigning SNP ancestry to representative samples from ISC2/3/4/5/6/7. Each column shows the percentage of SNPs assigned to each percentage probability of being donated from each population (eg ISC2) for a particular sample (eg BHU581/0). If many SNPs had high probabilities of being donated from a particular population, this implied that sample shared a recent ancestor with that population. BHU581/0 was a putative ISC5/6/7 hybrid, and had 9.4% of its SNPs with ancestry from ISC6 or ISC7, 7.2% from ISC7, and 10.9% from ISC5. In contrast, BPK275/0cl18 was the representative sample for ISC5, and 44.3% of its SNPs had confident ancestry from ISC5. No SNPs from any comparison were assigned probabilities between 50–95%, hence these rows are omitted from the table. These ancestry patterns indicated that BPK035/0cl1-BPK043/0cl2, BHU1011/0-BHU770/0cl1, BPK280/0-BPK615/0 and BPK158/0cl9 represented ISC4-related isolates but not mixtures. BHU774/0 and BHU744/0, showed evidence of both ISC6- and ISC7-derived segments. The six ungrouped samples (BHU274/0, BHU815/0, BHU764/0cl1, BHU574cl4, BHU572cl3 and BHU581cl2) were sampled between February 2007 and February 2010 and had high numbers of heterozygous SNPs. High heterozygosity could be symptomatic of genetically distinct parents in a hybrid. For example, if BHU572cl3 and BHU581cl2 (both May 2009) were mixes of ISC5 (BHU573/0, May 2009) and ISC7 (BHU200/0, July 2006) then only 16 out of 280 genotypes (BHU572cl3) and 15 out of 278 genotypes (BHU581cl2) would be unexplained. This was evaluated for different haplotype sharing options and matched the timing of samples (BHU572cl3, BHU574cl4 and BHU581cl2 in relation to BHU573/0; BHU815/0 in relation to BHU816/0). SNPs assigned to populations with a high probability in the 191x6 model provided a framework for predicting strain membership as well as admixture. Comparing the relative difference in distributions of homozygous and heterozygous SNPs between candidate parents within the sample set was sufficient to identify shared ancestry where related lineages were sampled, and visible in the aligned genotypes of the six strains with segments of both ISC5 and ISC6 ancestry. (n) Percentage of haplotype segment copies shared within and between groups. Recipients are shown as columns and donors as rows. Groups with complex ancestries received more segments than they donated: ISC8 from ISC5, ISC6 and ISC7; ISC3, ISC9 and ISC10 from ISC5. These 'recipient' groups were also associated with long external branch lengths in phylogenies. The expected probability per SNP in each population for the 191x191 co-ancestry analysis indicated that strains assigned to populations by Structure were assigned the same one with FineStructure (ISC2/3/4/5/6/7): ISC2 (copying probability >99%), ISC3 (>97%), ISC4 (>98%), ISC5 (>99%), ISC6 (>90%) and ISC7 (>97%). Genetically heterogeneous groups showed a higher rate of segments were received than donated from specific populations: ISC8 from ISC5 (11.3% vs 1.7%), ISC6 (16.6% vs 4.7%) and ISC7 (16.5% vs 6.2%). ISC9 (10.0% vs 0.8%) and ISC10 also received more from ISC5 (11.1% vs 2.5%). ISC5 also donated more to ISC3 (11.0% v 1.9%) suggesting possible ancestral genetic exchange, which may be associated with the

long branch lengths leading to ISC3. (o) Four-gamete test for recombination. Four gamete tests showed evidence of recombination in the core population and ISC1. 2043 from 117,055 (1.75%) SNP pairs had a recombination signature in the core population. For the subset of 2,015 SNPs with no missing data R=0.7/Mb. The 8 hybrids were BHU815/0, BHU764A1, BHU274/0, BHU574cl4, BHU581cl2, BHU572cl3 (all ISC5+ISC6/7), BHU744/0 and BHU774/0 (both ISC6/7 only). Recombination was evident at 25% of SNPs in ISC1 (184,827 out of 739,581 SNP pairs) based on the four gamete test. This signature varied over 13-fold from chr31 (54.2% of pairs) to chr23 (4%). (p) SNP and indel association with $Sb^V$ resistance. Fisher exact tests evaluated the association of $Sb^V$-resistance with SNPs (coded as the number of derived mutations: 0, 1 or 2) and indels (coded as the diploid number of inserted or deleted (-) basepairs). (q) SNP association with in-vitro SSG phenotype. SNPs with derived alleles in multiple ISC groups with 6+ non-reference genotypes for which the absolute difference in SSG-R and SSG-S allele frequency was >0.1 with the OR (odds ratio) >4 and Benjamini-Hochberg-Yekutieli corrected t-test p value <0.005. SNPs are ordered by chromosome and position. The SSG phenotypes were assessed as ln(EC50) values for the derived (D) and reference (R) alleles. To avoid arbitrarily categorising the sample in vitro $Sb^V$ phenotypes as R and S using a given threshold EC50 value, we compared the log-scaled EC50 values of each allele pair at the 57 SNPs identified using t-tests (p<0.005): the $Sb^V$-R alleles had log-scaled EC50 values of 1.13–1.84 compared to S ones of 0.01–0.50 (*Supplementary file 1*). Consequently, the 57 SNPs together have power to differentiate $Sb^V$-R and $Sb^V$-S phenotypes. (r) SNP association with clinical outcome from SSG treatment. SNPs with derived alleles in multiple ISC groups with 6+ non-reference genotypes for which the absolute difference in SSG-cure and SSG-failure allele frequency was >0.1 with the OR (odds ratio) >4. The SNPs are ordered by chromosome and position. The SSG phenotypes were assessed for the derived (D) and reference (R) alleles. We examined lines for which the patient was treated with SSG and was either cured (n=21) or treatment failed (n=14: death, relapse, no response to treatment) to test a null hypothesis that SSG outcome and SNPs were unlinked. An alternative hypothesis was that SSG-cure patients could be infected with $Sb^V$-S parasites, and SSG-fail with $Sb^V$-R (*Supplementary file 2*). 32 SNPs were associated with SSG cure (OR>4), including 11 nonsynonymous SNPs, but none at a significant level due to the small sample size. (s) Relative strength of purifying selection between ISC populations. Relative strength of purifying selection between ISC populations (columns 1 and 2) as ascertained as the relative rate of accumulation of derived alleles computed as R for all (top), nonsynonymous (middle) and synonymous (bottom) sites. We tested for differential selective pressures among the ISC groups where observed values of R statistics deviated by at least four standard errors (SE) from the expected value of 1 (Do et al 2015). The genome-wide values are shown along with the weighted block jackknife confidence intervals (lower, median, upper). Derived alleles were more frequent in ISC4 than ISC5, suggesting stronger purifying selection in ISC5. Comparing ISC2 (Bangladesh) to ISC3-7 (Nepal/India), we found a genome-wide (RG) excess of derived alleles had accumulated in the latter (RG=0.74, 6.15*SE). This was not directly attributable to selection because the difference was present in both nonsynonymous (RN=0.59, 4.42*SE) or synonymous sites (RS=0.66, 10.82*SE). ISC3/4/5/6 shared genetic ancestors dating to approximately the same origin, so a comparison of their rates should indicate variation in the relative accrual of derived alleles. Despite possessing the largest sample size (n=52) of all ISC populations, ISC5 gained significantly fewer derived alleles genome-wide compared to ISC3 (RG=0.75, 11.39*SE), ISC4 (RG=0.65, 7.37*SE) and to a lesser extent, ISC6 (RG=0.82, 3.38*SE). This trend was present in both nonsynonymous and synonymous SNPs but not significant for most comparisons (an exception was comparing synonymous changes in ISC5 vs ISC4 (RS=0.37, 7.80*SE). Similarly, the rate of accumulation of homozygous derived SNPs (R2 values) was higher in ISC4 relative to ISC5 (R2G=0.90, 3.87*SE). Antimonial in vitro resistance was highest in ISC5 (ln[EC50]=1.38 ± 1.45) and lowest in ISC4 (0.13 ± 1.12). Consequently, the relative difference in the rate of accumulation of derived alleles in ISC5 that was predominantly isolated in India may reflect different exposure to drug selection compared to ISC4 (mainly from Nepal). Further work is required to quantify this in relation to variable antimonial resistance levels among samples in each population. ISC7 was a genetic subgroup of ISC6 that may be a recent radiation or have undergone a substantial population bottleneck. No genome-wide difference was evident in terms of selction, but fewer derived synonymous SNPs were found in ISC6 (RS=2.18, 4.44*SE), suggestive stronger hitchhiking of alleles in ISC7. (t) Multi-allelic sites. Sites with multiple derived alleles

compared to reference genome sample BPK282/0cl4 in the core population (7, top) and ISC1 (10, bottom).

### Major datasets

The following dataset was generated:

| Author(s) | Year | Dataset title | Dataset URL | Database, license, and accessibility information |
|---|---|---|---|---|
| The Wellcome Trust Sanger Institute | 2015 | Genomic diversity in Leishmania donovani in Indian subcontinent | http://www.ebi.ac.uk/ena/data/view/ERP000140 | Publicly available at European Nucleotide Archive (accession no. ERP000140) |

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

**Appendix 1 - SNP detection procedure using the COCALL algorithm**

## 1 Consensus variant discovery to reduce tool-specific bias

COCALL (COnsensus of SNP CALL) was designed to overcome a lack of verified SNPs for *L. donovani* by combining SNP information from different SNP callers using a consensus calling scheme based on the framework outlined for the 1000 Genomes project (*1000 Genomes Project et al., 2012*). The main premise was that no SNP caller was *a priori* superior, and that errors among calling tools tended to differ across sites, whereas true variants showed a concordant signal. This approach avoided systematic errors unique to each individual SNP caller by examining their consistency and identifying discordant mutations symptomatic of false positives (*Weisenfeld et al., 2014*). These disparities among callers arise from their differing strategies for tackling repetitive regions, ambiguous read mapping probabilities and alignments, low complexity regions, tandem repeats, inversions, translocations and regions in which genuine repeats are collapsed in the reference assembly. By combining this with population-level information to scale the prior probability of a mutation existing at a known SNP site, COCALL is optimised for variant discovery in a largely unknown population. This integrated variant discovery scheme inferred sample genotypes both dependently and independently of other samples, which may mitigate bias in rare variant discovery: dependent approaches miss true SNPs and independent ones include many false positives (*Han et al., 2014*). COCALL was particularly effective when samples were genetically homogenous and the sequence quality and read depth among samples was uniform, as in this study. When these conditions were not met, COCALL was less effective, and population-based methods implemented in GATK, Freebayes or Samtools Mpileup performed better because these methods identified base variations based on aggregated depth across the population, and were less affected by lower quality sequence runs. Computational time for running COCALL may become prohibitively long for samples with high genetic diversity.

This scheme applied five different mutation detection approaches and computed the support for each base variation. COCALL interpreted the results of FreeBayes v0.9.5 (*Garrison and Marth, 2012*), GATK 2.0-38 (*McKenna et al., 2010*), Samtools Pileup v0.1.16 (*Li et al., 2009*) and Samtools Mpileup v0.1.18 based on reads mapped with Smalt v5.7, and also Cortex v1.0.5.13 (*Iqbal et al., 2012*) based on its own assembly and mapping (*Figure 2—figure supplement 1*).

## 2 Attributes of SNP callers used by COCALL

1) Cortex was used as a de novo assembly haplotype-based SNP caller and offered an independent assessment of SNPs because it did not rely on read alignment and mapping by Smalt. For SNP calling, we merged SNPs with scores $\geq 2.5$ identified on the current reference with de Bruijn graphs with k-mers of 31 and 63 bases; the latter were structurally more accurate at a cost of nucleotide level accuracy (*Iqbal et al., 2012*). This approach detected fewer SNPs overall than other methods, as expected (*Weisenfeld et al., 2014*). Cortex depended on de novo assemblies, so its accuracy was lower for repetitive regions and collapsed repeats. Cortex did not provide strand bias information, so the strand bias of Cortex SNPs was calculated from based on corresponding strand information from Pileup.

2) Freebayes applied a Bayesian strategy for SNP calling and was used to discover potential SNPs with SNP scores $\geq 50$ using default parameters for strand bias and read depth. Unlike other SNP callers, it detected variation based on the local haplotype across the read length, and so was superior for variant calling at regions with short repeats. It required disomic chromosomes: but for this study this limitation did not affect SNP calling because no strict additional parameters for SNP calling were applied to allow it identify rare variants.

3) GATK provided two SNP calling methods: one for individual samples, and one based on a population of related samples (GATK population caller). A SNP score threshold >50 was used for both. For our sample set, GATK had comparatively lower power to find SNPs at duplicated loci, and the base quality recalibration (BQR) scheme (*DePristo et al., 2011*) was not applicable due to the low number of known SNPs for training and the low total number of high-frequency candidate SNPs.

4) Samtools Mpileup was applied with a SNP score ≥20, and called fewer SNPs near indels compared to GATK and Freebayes. It assumed disomy, inferred SNPs across a population of samples, and ultimately had a lower false positive rate.

5) Samtools Pileup had a lower SNP detection power at regions with low read coverage, at gaps, and near indels. However, Pileup was more accurate at high-coverage regions with lower mapping quality and produced fewer false positives. Here, it was used for candidate SNPs with a base quality >20 and >10% of reads supporting a SNP being on both the forward and reverse strands to reduce strand bias which was calculated based on pileup read depth. 11.5% of all candidate SNPs had significant strand bias such that the proportion of either forward or reverse reads was >90%, and 4% had reads only for forward or for reverse strands.

# 3 Screening and consensus SNP calling in populations

Candidate SNPs with mapping qualities of ≥40 across all five callers were retained: approximately 52% of candidate SNPs were excluded due to low base, SNP or mapping quality values. SNPs at sites with normalised read depth ≤0.5 or depth ≥1.75 or with multiple derived alleles across the five callers were excluded. 1.75 was used as the upper read depth limit instead of 2.0, which is often usedas a criterion for duplication, because SNP callers estimated read depth based on base and mapping quality, so even true duplicated regions tended to have depth <2.0. In addition, SNPs present in clusters of more than three per seven bp window were removed as likely to be associated with tandem repeats, small indels and other sequence artefacts. The modal read depth coverage for each candidate SNP site across the core population was 49-fold, and the minimum modal read depth in any of the core population was 14-fold: 1,986,930 genotypes were computed in total.

A panel of 90 sequencing libraries from each of three well-characterised isolates (BPK282, BPK275 and BPK087) representing three major groups from our core population that were generated as part of this study were used to calibrate the SNP calling across the population. The metric of the mean number of callers per candidate SNP site was determined for the entire set of 300 libraries. Candidate SNP sites in lower complexity regions were accepted when the mean number of callers ≥3.5; those in non-masked regions were kept where the mean number of callers ≥2.5 (*Table A1.1*); those in hard-masked regions were omitted. These criteria resulted in a SNP set supported by a mean of 4.5 callers per SNP. An additional screening condition on the sum of the individual SNP caller score (>380) was applied to some lower quality runs (for isolates BHU569A1, BHU572A1, BHU574A1, BHU575A1, BHU581A1, BHU592A1, BHU741A1, BHU764A1, BHU770A1, BHU777A1, BHU782A1, BHU814A1, BPK031A1, BPK091A1, BPK157A1, BPK164A1, BPK406A1 and BPK512A1).

**Table A1.1.** Number of putative SNPs detected by different numbers of SNP calling tools. A threshold mean number of callers ≥3 was used, which was elevated to ≥4 at soft-masked regions.

| #SNP callers | Count | Cumulative count | Freq | Cumulative Freq |
| --- | --- | --- | --- | --- |
| 1 | 32,072 | 32,072 | 0.822 | 0.822 |
| 1.4 | 154 | 34,475 | 0.004 | 0.884 |
| 1.5 | 498 | 34,973 | 0.013 | 0.896 |

*Table A1.1 continued on next page*

*Table A1.1 continued*

| #SNP callers | Count | Cumulative count | Freq | Cumulative Freq |
|---|---|---|---|---|
| 2 | 965 | 36,160 | 0.025 | 0.927 |
| 2.5 | 38 | 36,244 | 0.001 | 0.929 |
| 3 | 476 | 36,757 | 0.012 | 0.942 |
| 3.5 | 114 | 36,935 | 0.003 | 0.947 |
| 4 | 1,019 | 38,046 | 0.026 | 0.975 |
| 4.5 | 179 | 38,347 | 0.005 | 0.983 |
| 5 | 438 | 39,015 | 0.011 | 1 |

The variant read allele frequency, read depth and sum of weighted SNP scores were measured for each putative SNP to evaluate SNP quality. While false positive SNPs would be expected to occur more frequently in clusters due to local structural variations or repetitive sequences: mean distance between COCALL SNPs was approximately equal between regions of the genome. Allele frequencies inferred from our genotype calls were highly correlated for the 2418 heterozygous SNPs ($r^2$=0.999) with those derived from frequencies of reads with each allele for the 191 core isolates. The heterozygous allele read depth distributions indicated some deviations from the expected value of 0.5 due to a mix of strand bias, chromosome copy number variation, and variation in read mapping and alignment. However, few SNPs were heterozygous and few genotypes were ambiguous: 14,825 mutations had read depth allele frequencies of 0.95 or greater.

# 4 Verification of variants called in individual libraries

We validated 15 true heterozygous SNPs in the reference BPK292/0cl4 isolate by manually inspecting mapped reads for this isolate, and confirmed that COCALL correctly called these sites. In addition, we examined variant calls on two isolates (BPK519/12 and BPK156/0; ISC1) where chromosome 17 is inferred to be monosomic. The mean derived allele frequency at variant sites on these chromosomes was 99.3% (at 1338 SNPs in BPK519/12) and 99.9% (at 1336 SNPs in BPK156/0). On this chromosome, COCALL called 1337 homozygous SNPs and a single heterozygous one for BPK519/12 and 1318 homozygous SNPs and 18 heterozygous ones for BPK156/0. While eliminating all heterozygous calls may seem optimal, it was possible that some cells were not monosomic for this chromosome, and choosing parameters that eliminated all heterozygous SNP calls removed many valid homozygous ones

# 5 Comparison of COCALL with GATK population calling

We tested the relative rate of true and false SNP sites and individual genotypes for COCALL and population-based GATK (*DePristo et al., 2011*) using read data of three previously studied isolates (*Downing et al., 2011*) (BPK282/0cl4, BPK087/0cl11, BPK275/0cl18, sampled from the most frequent genetic populations ISC4, ISC5 and ISC6 respectively). We created artificial pairwise mixes of these samples in proportions of 1:9, 2:8, 3:7, 4:6, 5:5, 6:4, 7:3, 8:2, 9:1 and unmixed, with depth levels adjusted for the relative proportions of unmapped reads in each. With these data, COCALL identified more rare variant SNPs in mixed libraries for the BPK275/0cl18-BPK087/0cl11 mix (*Table A1.2*). BPK282/0cl4 was the reference genome strain and so should possess only heterozygous SNPs, and so its variant discovery rate was substantially lower when combined in mixed libraries. Consequently, for the mixed libraries of BPK282/0cl4-BPK087/0cl11 and BPK282/0cl4-BPK275/0cl18, once the fraction of non-reference reads <30%, the sensitivity of COCALL was superior to that for the GATK population caller.

**Table A1.2.** The fraction of discordant true and false SNPs between COCALL and GATK population-based variant calling. COCALL had a greater proportion of true SNPs – confirmed by visual inspection of mapping reads – both in terms of the number of variant sites, and across the population of samples.

| SNP caller | Sites | | Mutations | |
| --- | --- | --- | --- | --- |
| | True | False | True | False |
| COCALL | 106 | 33 | 991 | 118 |
| | 76.3% | 23.7% | 89.4% | 10.6% |
| GATK population caller | 33 | 106 | 118 | 991 |
| | 23.7% | 76.3% | 10.6% | 89.4% |

Specifically, the population-based GATK and COCALL both called 11,389 variant genotypes in the artficially mixed samples, while GATK called 718 SNPs not detected using COCALL, and COCALL found 396 additional SNPs not in the GATK set: a total of 1109 discordant variants at 139 sites. Visual inspection of mapped reads using Artemis (*Carver et al., 2008*) and Samtools tview (*Li et al., 2009*) indicated that 89% (354) of these 396 unique to COCALL were valid. COCALL had a greater proportion of true SNPs both in terms of the number of variant sites, and across the population of samples. This illustrated COCALL's improved power for detecting variants at novel sites, compared with GATK (*Table A1.3*).

**Table A1.3.** The rate of SNP calling for different combinations of read libraries of samples BPK282/0cl4, BPK087/0cl11 and BPK275/0cl18. BPK282/0cl4 was the reference genome strain and so should possess only heterozygous SNPs, and so its variant discovery rate was substantially lower. COCALL identified more rare variant SNPs in mixed libraries.

| Read library combinations | | | | | |
| --- | --- | --- | --- | --- | --- |
| BPK282/0cl4 | BPK275/0cl18 | BPK087/0cl11 | Unique to COCALL | Shared | Unique to GATK population caller |
| | 89% | 11% | 14 | 140 | 6 |
| | 80% | 20% | 7 | 182 | 7 |
| | 70% | 30% | 6 | 191 | 7 |
| | 60% | 40% | 9 | 193 | 7 |
| | 50% | 50% | 7 | 194 | 7 |
| | 40% | 60% | 5 | 196 | 6 |
| | 30% | 70% | 7 | 191 | 6 |
| | 20% | 80% | 7 | 178 | 8 |
| | 11% | 89% | 9 | 145 | 6 |
| 89% | | 11% | 10 | 50 | 5 |

*Table A1.3 continued on next page*

*Table A1.3 continued*

| Read library combinations | | | | | Unique to GATK |
| BPK282/0cl4 | BPK275/0cl18 | BPK087/0cl11 | Unique to COCALL | Shared | population caller |
|---|---|---|---|---|---|
| 80% | | 20% | 11 | 112 | 5 |
| 70% | | 30% | 5 | 129 | 4 |
| 60% | | 40% | 7 | 131 | 5 |
| 50% | | 50% | 7 | 131 | 6 |
| 40% | | 60% | 6 | 132 | 6 |
| 30% | | 70% | 5 | 133 | 6 |
| 20% | | 80% | 4 | 130 | 6 |
| 11% | | 89% | 4 | 127 | 6 |
| 89% | 11% | | 15 | 64 | 3 |
| 80% | 20% | | 14 | 107 | 4 |
| 70% | 30% | | 4 | 129 | 3 |
| 60% | 40% | | 4 | 132 | 3 |
| 50% | 50% | | 3 | 133 | 4 |
| 40% | 60% | | 3 | 132 | 5 |
| 30% | 70% | | 4 | 131 | 4 |
| 20% | 80% | | 3 | 130 | 5 |
| 11% | 89% | | 3 | 126 | 6 |

COCALL discovered a total of 2418 SNPs (4.3% of the original candidate set) in the 191 isolates from the core population. We calculated that our calling approach had a 99% chance of finding variants segregating at a frequency $\geq$0.024 in the Core 191 and $\geq$0.173 in ISC1 (*Table A1.4*).

**Table A1.4.** The probabilities of observing alleles at a specific frequency in the core population and ISC1. We assessed whether or not the SNPs we discovered in our sample of ISC *L. donovani* were likely to be representative of the variation existing in the wider population of parasites circulating in the ISC. We first assessed the chance of discovering alleles at frequency f as $1-(1-f)2n$ in the core population (n=191) and ISC1 (n=12) sample sets (*Gutenkunst et al., 2009*). As a second approach, we evaluated the chance of sampling at least one derived allele in n sampled chromosomes (as above) assuming these come from a population of size N=1000 following a hypergeometric distribution (*Tennessen et al., 2012*). Both approaches confirm we have good power (>99% probability) of sampling alleles with population frequencies as low as 1.2% in the core population, but are likely to miss many variants in ISC1.

| | P(derived allele)=1-(1-f)$^{2n}$ | | $\binom{M}{0}\binom{N-M}{n}/\binom{N}{n}$ | |
| | | | P(derived allele)=1 – | |
| Allele Frequency | Core population | ISC1 | Core population | ISC1 |
|---|---|---|---|---|
| 0.001 | 0.318 | 0.024 | 0.382 | 0.024 |
| 0.002 | 0.535 | 0.047 | 0.618 | 0.047 |
| 0.003 | 0.683 | 0.07 | 0.91 | 0.07 |
| 0.004 | 0.784 | 0.092 | 0.945 | 0.092 |
| 0.005 | 0.853 | 0.113 | 0.966 | 0.115 |
| 0.006 | 0.9 | 0.134 | 0.979 | 0.136 |

*Table A1.4 continued on next page*

*Table A1.4 continued*

| | P(derived allele)=1-(1-f)$^{2n}$ | | $\binom{M}{0}\binom{N-M}{n}/\binom{N}{n}$ P(derived allele)=1 – | |
|---|---|---|---|---|
| Allele Frequency | Core population | ISC1 | Core population | ISC1 |
| 0.007 | 0.932 | 0.155 | 0.987 | 0.157 |
| 0.008 | 0.953 | 0.175 | 0.992 | 0.177 |
| 0.009 | 0.968 | 0.195 | 0.997 | 0.197 |
| 0.01 | 0.978 | 0.214 | 0.999 | 0.217 |
| 0.012 | 0.99 | 0.252 | 0.999 | 0.254 |
| 0.015 | 0.997 | 0.304 | 0.999 | 0.307 |
| 0.021 | 0.999 | 0.399 | 0.999 | 0.403 |
| 0.024 | 0.999 | 0.442 | 1 | 0.446 |
| 0.029 | 0.999 | 0.507 | 1 | 0.511 |
| 0.038 | 0.999 | 0.605 | 1 | 0.61 |
| 0.039 | 0.999 | 0.615 | 1 | 0.619 |
| 0.04 | 1 | 0.625 | 1 | 0.629 |
| 0.049 | 1 | 0.701 | 1 | 0.705 |
| 0.065 | 1 | 0.801 | 1 | 0.805 |
| 0.092 | 1 | 0.901 | 1 | 0.904 |
| 0.173 | 1 | 0.99 | 1 | 0.99 |
| 0.272 | 1 | 1 | 1 | 1 |

# 6 Singleton SNPs and transition-transversion ratios

Singleton SNP calls (those called in only a single isolate) are expected to be more error-prone than shared variants (*Han et al., 2014*). We examined the fractions of pairs of singletons; doubletons (SNPs observed in two samples); and all other SNPs ($\geq$3 samples) that co-occur within blocks of a range of sizes (10 b, 100 b, 500 b, 1 kb, 5 kb, 10 kb, 50 kb, 100 kb, 500 kb, 1 Mb, 5 Mb) to evaluate evidence of clustering: erroneous regions tend to have SNPs clustered together, and so more pairs of SNPs will co-occur within small blocks. We see this effect in SNP calls from Samtools Pileup (*Table A1.5*) but not for other callers or in the COCALL consensus. No difference between rates of SNPs called at coding and non-coding regions were observed. Very few sites were called as having more than two alleles (*Supplementary file 2, table t*).

**Table A1.5.** The percentage of SNP pairs for all pairs on the same chromosome for which both SNPs occur within a block, for different block sizes (kb) for Cortex, GATK, Samtools Mpileup and Pileup (Supplementary Table 6). Singletons may have clustered for Samtools Pileup, but not for Cortex, GATK or Samtools Mpileup. The mean distance was calculated the average distance between SNPs of the given type on the same chromosome. Non-rare SNPs were those found in three or more samples.

| | | Block size (kb) | | | | | | | | |
|---|---|---|---|---|---|---|---|---|---|---|
| SNP Caller | SNP type | 0.01 | 0.1 | 0.5 | 1 | 5 | 10 | 50 | 100 | 500 |

*Table A1.5 continued on next page*

*Table A1.5 continued*

| SNP Caller | SNP type | Block size (kb) | | | | | | | | |
|---|---|---|---|---|---|---|---|---|---|---|
| | | 0.01 | 0.1 | 0.5 | 1 | 5 | 10 | 50 | 100 | 500 |
| Cortex | Non-rare | 0.0 | 0.3 | 0.9 | 1.1 | 1.7 | 3.4 | 9.6 | 17.0 | 61.5 |
| | Doubletons | 1.5 | 1.5 | 1.5 | 1.5 | 3.1 | 3.1 | 10.8 | 13.8 | 64.6 |
| | Singletons | 0.0 | 0.0 | 0.0 | 0.0 | 3.8 | 3.8 | 23.1 | 30.8 | 80.8 |
| | All | 0.0 | 0.3 | 0.9 | 1.1 | 1.7 | 3.3 | 9.6 | 17.0 | 61.5 |
| GATK | Non-rare | 0.0 | 0.1 | 0.2 | 0.3 | 1.1 | 2.1 | 9.3 | 17.8 | 64.4 |
| | Doubletons | 1.3 | 1.3 | 1.3 | 1.3 | 1.3 | 1.3 | 5.3 | 13.3 | 61.3 |
| | Singletons | 0.0 | 0.0 | 0.0 | 0.0 | 1.1 | 3.2 | 11.8 | 18.3 | 77.4 |
| | All | 0.0 | 0.1 | 0.2 | 0.3 | 1.1 | 2.1 | 9.3 | 17.8 | 64.4 |
| Samtools Mpileup | Non-rare | 0.0 | 0.1 | 0.3 | 0.4 | 1.2 | 2.2 | 9.5 | 17.9 | 64.4 |
| | Doubletons | 0.0 | 0.0 | 0.0 | 0.0 | 6.7 | 6.7 | 6.7 | 20.0 | 60.0 |
| | Singletons | 0.0 | 0.0 | 0.0 | 0.0 | 9.1 | 9.1 | 9.1 | 9.1 | 81.8 |
| | All | 0.0 | 0.1 | 0.3 | 0.4 | 1.2 | 2.2 | 9.5 | 17.9 | 64.4 |
| Samtools Pileup | Non-rare | 0.0 | 0.1 | 0.2 | 0.3 | 1.1 | 2.1 | 9.3 | 17.8 | 64.4 |
| | Doubletons | 2.6 | 3.3 | 3.3 | 3.3 | 5.3 | 6.6 | 11.8 | 21.1 | 63.2 |
| | Singletons | 0.8 | 4.4 | 5.2 | 5.2 | 5.7 | 6.3 | 13.4 | 24.9 | 71.6 |
| | All | 0.0 | 0.1 | 0.2 | 0.3 | 1.1 | 2.1 | 9.3 | 17.8 | 64.5 |

We also looked at the transition/transversion (Ti/Tv) ratio, as transitions are expected biologically to be more common than transversions, while random (noisy) SNP calls will have Ti/Tv close to 0.5, as there are 4 possible transition mutations and 8 transversions. We calculated Ti/Tv ratios for variants in the core population of 191 (plus one additional closely related sample, BHU1087/0). We find that SNP calls supported by multiple callers had higher Ti/Tv ratios (*Table A1.6*), with the highest ratio for SNPs called by all five callers (2.74). Ti/Tv ratios consistently differed between singletons, doubletons and non-rare variants regardless of SNP caller: for SNPs called by all five methods, non-rare variants had higher rates (Ti/Tv=7267/1914=3.80) than doubletons (558/298=1.87) or singletons (1316/1124=1.17), as expected if singleton variants are more likely to be false positives. Ti/Tv ratios of non-rare SNPs were much higher than singleton SNPs across SNP sets called by Cortex-Mpileup (3.61 vs 0.91), Cortex-Pileup (3.08 vs 0.79), Cortex-GATK (2.26 vs 0.53), and Cortex-Freebayes (1.51 vs 0.40). While non-protozoan inter-species comparisons find a Ti/Tv ratio of 2.0–2.1 for genome-wide datasets and 3.0–3.3 for coding SNPs (*Ebersberger et al., 2002*; *Freudenberg-Hua et al., 2003*), we find higher Ti/Tv ratios at non-coding regions than coding regions, irrespective of SNP caller(s) used. Moreover, this effect was higher for high-confidence SNPs such as those called by all five callers and those present in multiple isolates. For example, for SNPs called by all five calling tools, all variants: coding Ti/Tv=4154/1736=2.39 vs non-coding 4987/1600=3.12; non-rare SNPs: coding 3285/1033=3.18 vs non-coding 3982/881=4.52; singleton SNPs: coding 617/565=1.09 vs non-coding 699/559=1.15.

**Table A1.6.** The transition (Ti) and transversion (Tv) mutation rates and Ti/Tv ratio across five SNP callers (Cortex, Freebayes, GATK, Samtools Mpileup, Samtools Pileup). SNP sets were selected from the intersection of the SNP calling methods marked X on the corresponding columns (for example, the first row uses the intersection of SNPs called by all five methods). The independent genome assembly and haplotype-based variant calling scheme utilised by Cortex complemented other calling methods based on Smalt assemblies well, based on the increase of the Ti/Tv ratio.

| Cortex | Freebayes | GATK | Samtools mpileup | Samtools pileup | Ti | Tv | Ti/Tv |
|---|---|---|---|---|---|---|---|
| X | X | X | X | X | 9141 | 3336 | 2.74 |
| X | | X | X | X | 9229 | 3411 | 2.71 |
| X | X | | X | X | 9150 | 3418 | 2.68 |
| X | | | X | X | 9239 | 3499 | 2.64 |
| X | X | X | X | | 9323 | 3832 | 2.43 |
| X | | X | X | | 9418 | 3951 | 2.38 |
| X | X | | X | | 9339 | 3960 | 2.36 |
| X | | | X | | 9436 | 4127 | 2.29 |
| X | X | X | | X | 9194 | 4426 | 2.08 |
| X | X | | | X | 9221 | 4591 | 2.01 |
| X | | X | | X | 9289 | 4628 | 2.01 |
| X | | | | X | 9342 | 4841 | 1.93 |
| X | X | X | | | 9507 | 6899 | 1.38 |
| X | | X | | | 9642 | 7536 | 1.28 |
| X | X | | | | 10,399 | 13,101 | 0.79 |
| X | | | | | 13,351 | 28,388 | 0.47 |
| | X | | | | 1,904,004 | 3,340,432 | 0.57 |
| | | X | | | 152226 | 553,889 | 0.27 |
| | | | X | | 41,107 | 137,315 | 0.30 |
| | | | | X | 42,902 | 153,295 | 0.28 |

# Appendix 2 - Somy, CNV, indel and episome detection

## 1 Chromosome copy number estimation

The reads mapped by Smalt v5.7 were used to estimate the read depth for each sample at each chromosome and region using the coverage values from Samtools v0.1.18. Chromosomal read depths were computed from the median read depth for sites with depths within one standard deviation of the initial estimate of the depth for each chromosome, and normalised as the depth per haploid genome as outlined previously (*Downing et al., 2011*). The genome-wide values per sample allowed the assignment of monosomic, disomic, trisomic, tetrasomic, pentasomic and hexasomic chromosomes as continuous values to allow for intra-sample variation in somy among cells. Where the most frequent somy was trisomy rather than disomy, then chromosome copy number was normalised to reflect that trisomy was the median chromosome read depth of a given strain. The range of monosomy, disomy, trisomy, tetrasomy and pentasomy was defined to be the haploid normalized chromosome depth $d_{ch}$ of $d_{ch} < 0.65$, $0.65 \leq d_{ch} < 1.25$, $1.25 \leq d_{ch} < 1.75$, $1.75 \leq d_{ch} < 2.25$ and $2.25 \leq d_{ch} < 2.75$ (*Table A2.1*).

**Table A2.1.** Somy status of 36 chromosomes in 206 strains (204 with JPCM5 and LV9). The range of monosomy, disomy, trisomy, tetrasomy and pentasomy was defined to be the haploid normalized chromosome depth $d_{ch}$ of $d_{ch} < 0.65$, $0.65 \leq d_{ch} < 1.25$, $1.25 \leq d_{ch} < 1.75$, $1.75 \leq d_{ch} < 2.25$ and $2.25 \leq d_{ch} < 2.75$.

|  | Count | Percentage |
| --- | --- | --- |
| Monosomy | 2 | 0.03 |
| Disomy | 5851 | 78.90 |
| Trisomy | 1229 | 16.57 |
| Tetrasomy | 310 | 4.18 |
| Pentasomy | 24 | 0.32 |

## 2 Copy number variation

Baseline coverage levels were established using median depth of over 60 strains among ISC4, ISC6, ISC7 and ISC8 so that potential CNVs were determined in comparison to the haploid normalised depth. For CNV detection, we also considered CNV length along with depth because long heterozygous deletions or duplications were observed that could be reliably identified at lower depth than shorter variants. Therefore we used three different depth thresholds: for CNVs of 2 to 5 kb, the threshold was five times the standard deviation of the chromosomal depth; for CNVs of 5 to 20 kb, the threshold was three times this value; for CNVs of >20 kb, the threshold was 1.5 times this value.

Adjacent CNV pairs in individual samples were combined when two nearby CNVs were located within 1/3 of the length of the longer of the two CNVs, because two independent adjacent CNVs would be a less parsimonious pattern. When a CNV pairwise distance was calculated, common boundaries of a given CNV among the samples were assigned to include every start and end of the CNV of each strain, and then the depth was calculated for each CNV range. Boundaries of CNVs were manually checked and CNVs overlapping large gaps were excluded. Candidate CNVs in the first 300 bp and last 5 Kb of all chromosomes were excluded. Adjacent CNVs were merged if the inter-CNV gap was smaller than the average length of the flanking CNVs.

## 3 Insertion and deletion discovery

Small indels, less than 10 bp, were detected by a four-caller method (Freebayes, GATK, Samtools Mpileup, Cortex). Pileup was not used because its indel detection method was not sufficiently powerful. The indel calling scheme was analogous to COCALL except that indels identified by at least three methods were accepted (instead of 2.5). To count allele frequency of indels at AQP1 more precisely, we used Smalt v7.1 that can map reads around indels more accurately by locating indels at the same base position for both forward and reverse strands. The AQP1 indel at 7,739 on chromosome 31 was visually inspected for all the strains using IGV and Samtools tview.

## 4 MAPK1 and H-locus amplicon detection

Read depth coverage was used to identify potential extra- or intra-chromosomal gene amplifications where levels were significantly elevated over an extended 10–50 Kb regions flanked by repetitive segments. Two of these variants: the copy number variation of the MAPK1 and H-locus amplicons was verified in a qPCR assay (SensiMix SYBR No-ROX, Bioline) performed on the LightCycler 480 (Roche Life Science). The genes MRPA (forward 5'-TGTGTTTCCGACGATTGC-3', reverse 5'-GTGACCCGCTTTGTGGAC-3') and MAPK1 (forward 5'-GTGGTCGCGCTGCAGAAG-3', reverse 5'-CGGCACAACCCCTTCATTG-3') were targeted in a subset of 46 samples. Cysteine synthase (LdBPK_363750) was used as a reference gene (forward 5'-GTCTTGGCGGTTCAGTTCG-3', reverse 5'-GACATTGTGGTTCGTCTGCTC-3'), and BPK026/0cl5 was used for normalization of copy numbers. This strain has three and two chromosomal copies of the H-locus and MAPK1, respectively.

The nature of the amplification (extra- or intra-chromosomal) was determined by pulsed-field gel electrophoresis (PFGE) and southern blot hybridization. Intact chromosomes were prepared from logarithmic phase cultures of *Leishmania* promastigotes and separated by PFGE using a Bio-Rad CHEF-Mapper XA at 6V/cm, 120° separation angle and a range of separation from 100 kb to 2.5 Mb. Gels were transferred by capillarity and hybridized with [$\alpha$-$^{32}$P]dCTP-labelled DNA probes specific for MAPK1, HTBF (H-locus) or LinJ35.4130 (LdBPK_354130), according to standard protocols.

To exclude the possibility that the amplicons are a culturing artefact, a PCR using primers that enable amplification on circular episomes or tandem duplications was also attempted directly on five bone marrow samples from VL patients. The edges of the chromosomal loci that code for the MAPK1 and DHFR-TS genes are targeted with primers (MAPK1: forward 5'-CATGGCGCAGTGACCTTCAG-3', reverse 5'-TCTTGGCACGGCATCAGCAG-3'; DHFR-TS: forward 5'-CCTACCCGCTTGCTTGCTTG-3', reverse 5'- CAGCAGCACAATGGAAAGAACG-3').

## 5 Correlation between SNPs, CNVs, indels and somy variation

The correlation coefficients of pairwise distances between samples based on SNP, CNV, indel

and somy diversity were calculated. Some strains were clearly phylogenetically related, so correlation between these values was expected simply due to the phylogenetic structure of our samples. To alleviate this potential bias, we used Pearson's correlation for sets of samples with different degrees of similarity based on SNP pairwise distance. The correlation coefficients of SNPs with indels and CNVs with SNPs were consistently large in all sample sets (*Table A2.2*).

**Table A2.2.** Pearson's correlation ($r^2$) between SNP, CNV, indels and somy diversity. The dependency of the correlation on the normalised pairwise distance is shown: this was determined from the median normalised pairwise distance based on SNP data. The Mantel correlation tests showed values that were consistent with the correlations.

| Type | #samples | Normalised pairwise distance threshold | Correlation ($r^2$) | P value |
|---|---|---|---|---|
| SNP-INDEL | 191 | 0 | 0.433 | 0.00E+00 |
| | 57 | 0.15 | 0.301 | 2.91E-254 |
| | 33 | 0.3 | 0.271 | 8.23E-77 |
| | 191 | Mantel test | 0.420 | 0.001 |
| PLO-SNP | 191 | 0 | 0.073 | 0.00E+00 |
| | 57 | 0.15 | 0.152 | 2.69E-118 |
| | 33 | 0.3 | 0.304 | 9.62E-88 |
| | 191 | Mantel test | 0.060 | 0.001 |
| PLO-INDEL | 191 | 0 | 0.024 | 3.88E-197 |
| | 57 | 0.15 | 0.040 | 2.43E-30 |
| | 33 | 0.3 | 0.072 | 1.96E-19 |
| | 191 | Mantel test | 0.016 | 0.001 |
| CNV-SNP | 191 | 0 | 0.289 | 0.00E+00 |
| | 57 | 0.15 | 0.263 | 7.53E-218 |
| | 33 | 0.3 | 0.393 | 4.52E-120 |
| | 191 | Mantel test | 0.272 | 0.001 |
| CNV-PLO | 191 | 0 | 0.121 | 0.00E+00 |
| | 57 | 0.15 | 0.150 | 1.96E-116 |
| | 33 | 0.3 | 0.183 | 1.59E-49 |
| | 191 | Mantel test | 0.103 | 0.001 |
| CNV-INDEL | 191 | 0 | 0.099 | 0.00E+00 |
| | 57 | 0.15 | 0.087 | 2.34E-66 |
| | 33 | 0.3 | 0.108 | 1.04E-28 |
| | 191 | Mantel test | 0.084 | 0.001 |

Correlations of our values with somy were more variable, probably because somy was more prone to homoplasy as it appears to mutate faster than other genetic variants in this study (*Table A2.2*). Spearman's rank correlations were also calculated and showed a similar level of correlation (data not shown). We also used Mantel tests to directly test correlation between the full matrices of pairwise distances. The results of Mantel tests were in concordance with Pearson's correlation values. Pearson's and Spearman's correlations were calculated using python-stats-0.6 (Gary Strangman http://sourcecodebrowser.com/python-stats/0.6/stats_8py. html). Mantel tests were implemented in R package vegan (https://github.com/vegandevs/vegan).

# 6 Correlation between duplicated genes and somy variation

Information on duplicated genes in the reference genome was based on multi-copy gene families annotated in the *L. infantum* JPCM5 reference genome (*Table A2.3*) (*Rogers et al., 2011*). In the present study, each read was mapped to its most similar region in the reference, and where reads could be mapped with equal mapping scores to multiple locations, one position was selected randomly to quantify read depth, in contrast to the previous study in which read depth was under-estimated by only using uniquely-mapped reads (*Rogers et al., 2011*). A threshold of 1.5 times the haploid depth was used to identify duplicated genes (a more conservative threshold of 2.0 did not change results materially). This threshold of 1.5 was effective for Illumina Hiseq data, where fluctuations in read depth were smaller than those previously seen using the Illumina Genome Analyser II. To illustrate this, the normalised standard deviation (standard deviation divided by average chromosome depth) of 17 strains sequenced by Genome Analyser II in was 0.32 (*Downing et al., 2011*), and in the current analysis, using Illumina Hiseq 2000 data, this was 0.19. We were therefore able to quantify more subtle depth variation than previous analyses. To normalise for differences in chromosome lengths, the number of duplicated genes was expressed per unit chromosome with a length set to that of chromosome 1. Results from between-species analysis (*Rogers et al., 2011*) have previously suggested a link between somy variation and copy number variation. However, in the present study, the correlation of duplicated genes per chromosome with the somy level was statistically significant but the effect size was very small (n=7146 chromosomes, $r^2$=0.027, p=7.03x10$^{-46}$), suggesting no relationship or only a weak one, between somy level and copy number at the population level.

**Table A2.3.** Correlation between CNV and ploidy of chromosomes in 206 strains: the duplication threshold was 0.5. p values <0.02 were observed for chromosomes 1, 2, 3, 17, 19, 20, 21 and the whole genome.

|  | Correlation ($r^2$) | P value |
| --- | --- | --- |
| chromosome | 0.027 | 7.03E-46 |
| 1 | 0.039 | 4.23E-03 |
| 2 | 0.202 | 1.21E-11 |
| 3 | 0.074 | 7.81E-05 |
| 4 | 0.003 | 4.31E-01 |
| 5 | 0 | 9.30E-01 |
| 6 | 0.004 | 3.86E-01 |
| 7 | 0.003 | 4.68E-01 |
| 8 | 0.001 | 6.05E-01 |
| 9 | 0.001 | 7.32E-01 |
| 10 | 0.015 | 7.77E-02 |
| 11 | 0.002 | 4.88E-01 |
| 12 | 0.005 | 3.16E-01 |
| 13 | 0.001 | 6.57E-01 |
| 14 | 0.001 | 7.31E-01 |
| 15 | 0.001 | 7.42E-01 |
| 16 | 0.009 | 1.81E-01 |
| 17 | 0.036 | 6.06E-03 |
| 18 | 0.005 | 3.23E-01 |
| 19 | 0.077 | 5.41E-05 |
| 20 | 0.064 | 2.51E-04 |

*Table A2.3 continued on next page*

*Table A2.3 continued*

|  | Correlation ($r^2$) | P value |
|---|---|---|
| 21 | 0.027 | 1.87E-02 |
| 22 | 0.009 | 1.87E-01 |
| 23 | 0.007 | 2.42E-01 |
| 24 | 0 | 9.68E-01 |
| 25 | 0.009 | 1.87E-01 |
| 26 | 0.017 | 5.97E-02 |
| 27 | 0.009 | 1.85E-01 |
| 28 | 0.001 | 6.80E-01 |
| 29 | 0.009 | 1.67E-01 |
| 30 | 0.003 | 4.28E-01 |
| 31 | 0.004 | 3.77E-01 |
| 32 | 0.005 | 3.28E-01 |
| 33 | 0.011 | 1.41E-01 |
| 34 | 0.001 | 7.61E-01 |
| 35 | 0.011 | 1.35E-01 |
| 36 | 0.016 | 7.38E-02 |

