## [Decision Letter]

Thank you for submitting your work entitled "Evolutionary genomics of epidemic visceral leishmaniasis in the Indian subcontinent" for consideration by *eLife*. Your article has been reviewed by three peer reviewers, and the evaluation has been overseen by Dominique Soldati-Favre as the Reviewing Editor and Detlef Weigel as the Senior Editor.

The following individual involved in review of your submission has agreed to reveal their identity: Peter Myler (peer reviewer).

The reviewers have discussed the reviews with one another and the Reviewing Editor has drafted this decision to help you prepare a revised submission.

Summary:

This study constitutes an unprecedented and major effort between scientists and clinicians from Europe and the Indian sub-continent. It brings comprehensively together various threads of the studies on the genetic diversity of *Leishmania donovani* isolates, genetic evolution, clinical response to drugs and drug susceptibility and resistance. The merit of this work is both the in depth analysis and in the unique strain collection that was subjected to sequencing

The authors have elegantly shown the power of genomic epidemiology. They have also linked their analysis to antimonial resistance, a phenotype that is widespread in part of India and hence the findings are of considerable importance for our knowledge of visceral leishmaniasis (VL) in the India Nepal focus. While much of the underlying sequence data has been described elsewhere (Downing et al., 2011, 2012), the sophisticated analyses presented here on the genetic diversity and evolution is of great importance to understand the biology of these understand the biology of these parasites.

Essential revisions:

1) There is a need for update on the VL situation in India and Nepal, which is no longer an epidemic, rather a picture of declining numbers of patients (see Abstract and Introduction).

2) Abstract: there is little evidence that Sb resistance has driven the epidemic, as opposed to historic known cycles of diseases, when later (Results, sixth paragraph etc.) it is stated that there is no statistical association between SNPs distribution and drug resistance.

3) Discussion, last paragraph: the statement that this data provides basis for prospective study on emerging drug resistance and surveillance is not justified by the data or results presented.

4) In the subsection “Sample phenotyping”/[Supplementary-material SD1-data]: it is clear from the paper that this definition of resistance and susceptible is based upon that used in previous publications by the group. However, the use of a single concentration value, in this case the EC50 value, is not valid to define any difference between 2 populations. In the case of only a 3-fold difference, which could be within bounds of experimental variation, this even less valid.

5) As acknowledged by the authors, many resistant isolates (outside ISC5 group) do not have mutations in AQP1 and some sensitive isolates of the ISC5 group (but also one of the ungrouped category) also have SNPs in AQP1. Thus the correlation between AQP1 and susceptibility is not perfect, even if its stands as a strong candidate. Ideally this hypothesis would get a lot of support if the authors could either introduce a WT version of AQP1 in strain with SNPs (with CRISPR/CAS this would be possible) or possibly less technically challenging introduce a WT and mutated episomal version of AQP1 in a sensitive and resistant isolates and look at the phenotype.

6) As far as I know AQP1 is a transporter of SbIII (not SbV). SbV is used for treating *Leishmania*. While they are potential explanations, this needs to be explained.

7) How is this study comparing to their 2011 Genome Research paper? In other words have they missed at the time the AQP1 SNPs or all their resistant isolates have a different resistance mechanism?

8) In the sixth paragraph of the Results section they indicate that all 52 ISC5 strains has a SNP in AQP1. It is unfortunate that they do not have the data on susceptibility for a majority of these strains (this would have helped in linking SNPs and phenotype).

9) I believe that the authors should not discount the role of the increase in copy number of the H locus as a determinant of resistance to SbV. The H locus contains MRPA one of the main antimony resistance genes. This should be discussed in more details and can it help in explaining some of the outliers?

10) In Figure 3 they indicate 20-30 copy number of MAPK1 gene and indicate that this is intrachromosomal amplification. The data shown in the Southern blot (Figure 3—figure supplement 2) is not supporting an increase in copy number of 20-30. One would even expect a change in size of the chromosome if they were that many intrachromosomal amplification. A control probe would be helpful. A minor point in Figure 3—figure supplement 1 (they discuss about episome, rather than intrachromosomal amplification).

---

## [Author Response]

Essential revisions: 1) There is a need for update on the VL situation in India and Nepal, which is no longer an epidemic, rather a picture of declining numbers of patients (see Abstract and Introduction).

We agree, and have adapted the Abstract and introductory paragraph to reflect this changing clinical picture as follows:

In the Abstract, we have changed “A recent epidemic in the Indian subcontinent (ISC) caused up to 80% of global VL and over 30,000 deaths per year” to now read “A recent epidemic in the Indian subcontinent (ISC) caused almost 80% of the global VL cases and over 30,000 deaths per year”.

In the Introduction, we have added a sentence and reference describing the recent decline in cases: “Recent intensified control efforts have led to a notable decline in cases (Chowdhury et al. 2014) but the problem is not yet eliminated”.

*2) Abstract*: *there is little evidence that Sb resistance has driven the epidemic, as opposed to historic known cycles of diseases, when later (Results, sixth paragraph etc.) it is stated that there is no statistical association between SNPs distribution and drug resistance.*

We agree this was overstated and have re-worded this as:”Resistance against antimonial drugs has probably been a contributing factor in the persistence of this epidemic”.

*3) Discussion, last paragraph*: *the statement that this data provides basis for prospective study on emerging drug resistance and surveillance is not justified by the data or results presented.*

We partially agree – our sentence stated that these data provide ‘the background’ for future prospective study, which we think is quite different from providing the ‘basis’ for such a study – obviously there are many other factors necessary for such work. However, the current formulation has clearly misled at least one reader, and we have reformulated this to now read:

”The data we present here provide baseline information on the diversity of *Leishmania donovani* in the ISC that will contribute to future studies of drug resistance and epidemiology of this population. Our results show the promise of genomic surveillance for other *Leishmania* populations, where patient symptoms, the parasites involved and the main treatment modalities all differ from those in the ISC (Sundar & Chakravarty, 2015).”

We hope this is more specific and accurate, and at least reasonably uncontroversial as now written.

*4) In the subsection “Sample phenotyping”/[Supplementary-material SD1-data]*: *it is clear from the paper that this definition of resistance and susceptible is based upon that used in previous publications by the group. However, the use of a single concentration value, in this case the EC50 value, is not valid to define any difference between 2 populations. In the case of only a 3-fold difference, which could be within bounds of experimental variation, this even less valid.*

We agree that EC50 values alone are not sufficient to define resistance and susceptibility. To make this clearer, we have added an additional column to [Supplementary-material SD1-data] showing the Activitiy Indices (AI) we used to define SSG susceptibility. These AI are a normalized way to express in vitro drug susceptibility, based on the EC50 value of that strain divided by the EC50 of a reference sensitive strain used in each assay (see Rijal et al., 2007, Microbes Infect 9: 529-35), and do not vary continuously between tested isolates, but are rather bimodal, with most strains showing an AI ≤1 (thus with an EC50 similar or lower than that of a reference sensitive strain) or ≥6 (thus with an EC50 6 times higher than the one of the reference sensitive strain). A few (7) strains show intermediate values (all around 3 or 4), many of which are explained as being the hybrids between R and S strains described in the paper. These AI values thus allow us to classify most isolates into ‘resistant’ and ‘sensitive’ classes.

This should have been explained more fully in the manuscript. We have amended the table and added to the text as follows:

“The classification into resistance and susceptible strains was determined by calculating the activity index (AI), i.e. the ratio of the EC50 of the strain in question versus the EC50 of the susceptible reference strain. AI values clustered strongly, with most strains showing an AI ≤1 (25; classified as SSG-sensitive) or ≥6 (18; classified as SSG- resistant). A few strains (7) showed AI values around 3 and were considered as showing intermediate resistance.”

*5) As acknowledged by the authors, many resistant isolates (outside ISC5 group) do not have mutations in AQP1 and some sensitive isolates of the ISC5 group (but also one of the ungrouped category) also have SNPs in AQP1. Thus the correlation between AQP1 and susceptibility is not perfect, even if its stands as a strong candidate. Ideally this hypothesis would get a lot of support if the authors could either introduce a WT version of AQP1 in strain with SNPs (with CRISPR/CAS this would be possible) or possibly less technically challenging introduce a WT and mutated episomal version of AQP1 in a sensitive and resistant isolates and look at the phenotype.*

We agree, of course, that this experimental confirmation would be ideal but we feel that this is beyond the scope of the population genetic analysis we have performed here and represents a substantial additional body of work, and – as the reviewer acknowledges – that the AQP1 deletion is a strong candidate without this validation. We argue that the originality of our study in the context of drug resistance is to show that a frameshift in this known SSG-R associated locus is fixed in a major subgroup of the ISC *L. donovani* population, and heterozygous in some hybrids with intermediate drug resistance phenotypes.

While the CRISPR-Cas9 system has been employed for *Leishmania donovani*, only two laboratories have, to our knowledge, published successful results with this technique in *Leishmania*. Many other groups are no doubt attempting to establish this approach, but we note that this is still some years from being a routine tool in this genus: for example, none of the groups contributing to this paper have experience with these tools in *Leishmania*. These approaches are likely to be made more challenging by the tetrasomy of chromosome 31 that carries the AQP1 locus. While episomal expression of different AQP1 alleles is likely to be more tractable, this approach is less likely to be completely definitive, and similar experiments have been performed by other labs:

Plourde et al., 2015, MBP: “The AQP1-null mutant was resistant to antimonyl tartrate (SbIII) and arsenite (AsIII) due to a decrease import of these metalloids.”

Monte-Neto et al., 2015, PLOS-NTD: “All mutants also displayed a reduced accumulation of antimony mainly due to genomic alterations at the level of the subtelomeric region of chromosome 31 harboring the gene coding for the aquaglyceroporin 1”.

Mandal et al., 2010, JAC: “Transfection of the AQP1 gene in a sodium antimony gluconate-resistant field isolate conferred susceptibility to the resistant isolate*”.*

We already mention some of this work, but now include a reference to the Plourde et al. paper to further emphasise this evidence, adding the sentence:

“Recently, an AQP1 knockout line of *Leishmania* major was shown to be resistant to Sb^III^ due to reduced uptake (Plourde et al. 2015).”

It is true that some resistant isolates do not show the indel in AQP1, and we are careful not to claim that this mutation is responsible for all resistance to antimonial drugs in this population. For example, a previous study on parasites we assign to the ISC4 group (with a wild-type AQP1 locus) has shown that these have decreased transcription of AQP1 (BPK087 and BPK190; Decuypere et al., 2005, AAC 49: 4616-21), and one isolate (BHU764, which is not in any of the main groups) has a different indel mutation in AQP1, and shows low mRNA expression of this locus, but also shows high expression of MRPA, an efflux transporter of SbIII (Mukhopadhyay et al., 2011. Int J Parasitol. 41:1311-21). We have added a sentence to emphasise this point, which we already make in the paper (Results, sixth paragraph):

”Indeed, other Sb^V^ resistance mechanisms are known in this population: previous work has shown that two resistant strains from ISC4 (BPK087 and BPK190) show significantly decreased transcription of an AQP1 locus encoding a wildtype protein sequence (Decuypere et al., 2005), and BHU764 combines a different indel mutation in AQP1 and reduced expression of MRPA, an efflux transporter of Sb^III^ (Mukhopadhyay et al., 2011).”

*6) As far as I know AQP1 is a transporter of SbIII (not SbV). SbV is used for treating Leishmania. While they are potential explanations, this needs to be explained.*

We agree, and this should have been explained for the reader. We have now added a sentence explaining the link between SbV and SbIII in the sixth paragraph of the Results section:

“While recent antimonial drugs such as SSG are compounds of pentavalent antimony (Sb^V^), Sb^V^ is thought to act mostly as a pro-drug, being reduced to SbIII in both the macrophage phagolysosome (Frézard et al., 2001) and in the parasite itself (Denton et al., 2004; Decuypere et al., 2012).”

We also noticed in this context that we don’t define SSG in the manuscript, and have added a few words to address this oversight:

“[…] initially with quinine, then with the trivalent antimonial Sb^III^ (1915) and compounds of the less toxic pentavalent Sb^V^ (1922) such as sodium stibogluconate (SSG)”

*7) How is this study comparing to their 2011 Genome Research paper? In other words have they missed at the time the AQP1 SNPs or all their resistant isolates have a different resistance mechanism?*

We think there are a number of reasons why the AQP1 two-base indel was not detected at the time we wrote the 2011 Genome Research paper. Sequence data from 5 ISC5 isolates (all resistant) were available at that time, and these were sequenced using a different technology – using Illumina Genome Analyzer II (GAII) technology rather than the HiSeq platform available now. These platforms differ in read length (76-base reads rather than 100-base reads) and differ in both the polymerase error rate and sequencing throughput, reflected in lower read coverage available for these isolates in 2011. In addition, the variant detection algorithms developed at that time (pre-2011) were focused on accurate SNP-calling, and we used an off-the-shelf approach implemented in the samtools package for identifying indels that are now thought to be prone to generating ambiguous putative indels as the now standard approach of locally re-aligning reads near indel positions was not established at that time. Our new study directly addresses this limitation by applying an independent de novo assembler (Cortex) to improve indel detection and verification by local read re-alignment. In addition, it applies a superior read-mapping approach with Smalt that has a much reduced error rate compared to the 2011 implementation with Ssaha2. We also augmented the Cortex-based approach with indel calling for individual libraries with Samtools Mpileup, for populations with GATK, and using a distinct alignment algorithm with FreeBayes. The combination of many more ISC5 samples and better sequencing and analysis technology has allowed us to detect this very small indel.

*8) In the sixth paragraph of the Results section they indicate that all 52 ISC5 strains has a SNP in AQP1. It is unfortunate that they do not have the data on susceptibility for a majority of these strains (this would have helped in linking SNPs and phenotype).*

In vitro susceptibility tests are time-consuming in *Leishmania donovani* and particularly difficult to apply on clinical isolates as the in vitro growth behaviour of isolates differs. High quality and standardized phenotyping of 50 isolates represents a very significant effort by several laboratories involved in this paper. We note that the number of parasites we phenotype is enough to establish that the distribution of resistance is non-random across population groups in our data. For example, out of ISC5, we tested 11 lines and 9 of them were highly resistant to antimonials (82%); the other group in which we tested the susceptibility in a large set of isolates was ISC4, where we tested 15 isolates and only found 4 resistant (26%); this difference is statistically significant (p=0.0154, Fisher’s exact test two-tailed). As we now note in the paper (Edits in response to #5 above) 2 of the resistant isolates found in ISC4 show decreased AQP1 expression with a wild-type amino acid sequence.

*9) I believe that the authors should not discount the role of the increase in copy number of the H locus as a determinant of resistance to SbV. The H locus contains MRPA one of the main antimony resistance genes. This should be discussed in more details and can it help in explaining some of the outliers?*

We agree this deserved more emphasis, and have added a sentence to reflect this and cite the appropriate evidence:

“The H-locus includes MRPA, a gene involved in the efflux of Sb^III^ and associated with drug resistance (Leprohon et al., 2009).”

We have followed the reviewer’s suggestion and MRPA gene copy number does not appear to explain cases of antimonial resistance outside ISC5. This is in contrast to AQP1, where we see changes in resistant isolates in other groups. We also note that the homozygous frameshift in AQP1 will clearly render this locus non-functional, while the effect of changes in gene dosage are harder to predict. We feel that to really assess the impact of MRPA dosage we would need to investigate both copy number and gene expression in the same samples, preferentially directly in the amastigote stage.

10) In Figure 3 they indicate 20-30 copy number of MAPK1 gene and indicate that this is intrachromosomal amplification. The data shown in the Southern blot (Figure 3—figure supplement 2) is not supporting an increase in copy number of 20-30. One would even expect a change in size of the chromosome if they were that many intrachromosomal amplification. A control probe would be helpful. A minor point in Figure 3—figure supplement 1 (they discuss about episome, rather than intrachromosomal amplification).

This experiment was not intended to quantify the amplicon, but merely to determine whether the amplification is extra- or intra-chromosomal. If the amplicon was extra-chromosomal, on an episome, we would expect to see a second, much lower band hybridizing with the MPK1 probe, as in panel D of the same figure. We agree that this kind of Southern blot experiment (without controls) is not quantitative, but we are not concerned that these results are in contradiction to our estimated increase in copy number. The MPK1 locus is approximately 15 kb, so the 15 repeats per haploid genome we estimate in BPK282 (around 15) would correspond to around 225 kb, only 8% the length of the 2.7 Mb chromosome 36. In the gels shown in panels A and B, this large chromosome migrates to the so-called compression zone of OFAGE gels where the largest chromosomes co-migrate and where it would be impossible to observe even fairly substantial differences in chromosome size. We think copy number is estimated both far more precisely and more accurately by looking at the depth of ilumina reads mapping to this feature than by looking at either the intensity of hybridization or lengths of molecules migrating on a gel.

We have corrected the figure legend to Figure 3—figure supplement 1 and thank the reviewer for noticing this error.